Applied and Environmental Science
# Microbial Metabolic Redundancy Is a Key Mechanism in a Sulfur-Rich Glacial Ecosystem

Christopher B. Trivedi,a* Blake W. Stamps,a* Graham E. Lau,b* Stephen E. Grasby,c Alexis S. Templeton,b John R. Speara

aDepartment of Civil and Environmental Engineering, Colorado School of Mines, Golden, Colorado, USA
bDepartment of Geological Sciences, University of Colorado Boulder, Boulder, Colorado, USA
cGeological Survey of Canada—Calgary, Natural Resources Canada, Calgary, Alberta, Canada

**ABSTRACT** Biological sulfur cycling in polar, low-temperature ecosystems is an understudied phenomenon in part due to difficulty of access and the dynamic nature of glacial environments. One such environment where sulfur cycling is known to play an important role in microbial metabolisms is located at Borup Fiord Pass (BFP) in the Canadian High Arctic. Here, transient springs emerge from ice near the terminus of a glacier, creating a large area of proglacial aufeis (spring-derived ice) that is often covered in bright yellow/white sulfur, sulfate, and carbonate mineral precipitates accompanied by a strong odor of hydrogen sulfide. Metagenomic sequencing of samples from multiple sites and of various sample types across the BFP glacial system produced 31 metagenome-assembled genomes (MAGs) that were queried for sulfur, nitrogen, and carbon cycling/metabolism genes. An abundance of sulfur cycling genes was widespread across the isolated MAGs and sample metagenomes taxonomically associated with the bacterial classes *Alphaproteobacteria* and *Gammaproteobacteria* and *Campylobacteria* (formerly the *Epsilonproteobacteria*). This corroborates previous research from BFP implicating *Campylobacteria* as the primary class responsible for sulfur oxidation; however, data reported here suggested putative sulfur oxidation by organisms in both the alphaproteobacterial and gammaproteobacterial classes that was not predicted by previous work. These findings indicate that in low-temperature, sulfur-based environments, functional redundancy may be a key mechanism that microorganisms use to enable coexistence whenever energy is limited and/or focused by redox chemistry.

**IMPORTANCE** A unique environment at Borup Fiord Pass is characterized by a sulfur-enriched glacial ecosystem in the low-temperature Canadian High Arctic. BFP represents one of the best terrestrial analog sites for studying icy, sulfur-rich worlds outside our own, such as Europa and Mars. The site also allows investigation of sulfur-based microbial metabolisms in cold environments here on Earth. Here, we report whole-genome sequencing data that suggest that sulfur cycling metabolisms at BFP are more widely used across bacterial taxa than predicted. From our analyses, the metabolic capability of sulfur oxidation among multiple community members appears likely due to functional redundancy present in their genomes. Functional redundancy, with respect to sulfur-oxidation at the BFP sulfur-ice environment, may indicate that this dynamic ecosystem hosts microorganisms that are able to use multiple sulfur electron donors alongside other metabolic pathways, including those for carbon and nitrogen.

**KEYWORDS** MAGs, functional redundancy, glacier, metabolic redundancy, metagenome-assembled genomes, microbial communities, sulfur, sulfur metabolism

Address correspondence to John R. Spear, jspear@mines.edu.

* Present address: Christopher B. Trivedi, Interface Geochemistry, GFZ German Research Centre for Geosciences, Helmholtz Centre Potsdam, Potsdam, Brandenburg, Germany; Blake W. Stamps, UES, Inc., Dayton, Ohio, USA; Graham E. Lau, Blue Marble Space Institute of Science, Seattle, Washington, USA.

Microbial metabolism of sulfur on Canadian High Arctic ice is functionally redundant; this allows for a mixed microbial community to thrive in response to geochemical energy challenges such as redox variation and the dominance of a single element, S. Cool!

Arctic and Antarctic ecosystems such as lakes (1), streams (2, 3), groundwater-derived ice (i.e., aufeis [spring-derived ice]; 4, 5), and saturated sediments (6) provide insight into geochemical cycling and microbial community dynamics in low-temperature environments. One such ecosystem, Borup Fiord Pass (BFP), located on

northern Ellesmere Island, Nunavut, Canadian High Arctic, contains a perennial sulfidic spring system discharging through glacial ice and forming supraglacial and proglacial ice deposits (aufeis) as well as extensive deposits of cryogenic carbonates, elemental sulfur, and gypsum (Fig. 1) (7). While such deposits are formed as thin, dispersed layers on the ice and snow surface, seasonal melting can also result in accumulations of precipitates into small mounds and melt-out ponds.

BFP is located at 81°01′N, 81°38′W (Fig. 1, inset) at the high point of a low, mostly ice-free, north-south-trending valley that cuts through the Krieger Mountains. The spring system of interest is approximately 210 to 240 m above sea level and occurs near the toe of two glaciers that flow down from the surrounding mountains and coalesce in the pass (7). The spring system is thought to be perennial in nature but discharges from different locations along the glacier from year to year (7–10). Due to a lack of winter observations, it is not known if spring flow is year-round or limited to the warmer spring and summer months; however, growth of large aufeis formations present in summer months suggests continued discharge during the winter. A lateral fault approximately 100 m south of the toe of the glacier has been implicated in the subsurface hydrology, in that this may be one of the areas where subsurface fluid flow is focused at the surface through thick permafrost (7, 11). Furthermore, organic matter in shales along this fault may be a source of carbon for subsurface microbial processes such as sulfate reduction (10).

Previous research on the BFP spring has detailed microbial activity, redox bioenergetics (9), and the biomineralization of elemental sulfur ($S^0$; 7, 10, 11) and the cryogenic carbonate vaterite (12). Wright et al. (9) produced one metagenome from a mound of elemental sulfur sampled at the site in 2012. From the metagenomic data, the authors determined the potential bioenergetics of microbial metabolisms and found that, at least in surface mineral deposits, sulfur oxidation was likely the dominant form of metabolism present among the *Campylobacteria*. An exhaustive 16S rRNA gene sequencing study conducted on samples collected in 2014 to 2017 revealed a diverse assemblage of both autotrophic and heterotrophic microorganisms in melt pools, aufeis, spring fluid, and surface mineral deposits that persisted over multiple years, contributing to a basal community present in the system regardless of site or material type (13). Despite the extensive taxonomic study, no work to date has attempted to fully characterize the metabolic capability of the microbial communities adapted to this glacial ecosystem beyond the surface precipitates at the site.

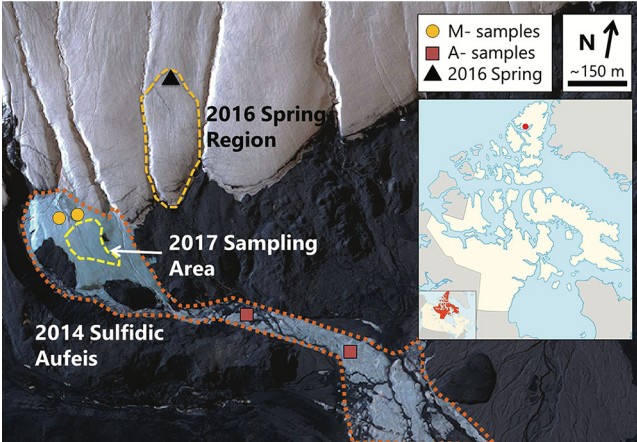

**FIG 1** Borup Fiord Pass satellite view. The image was collected on 2 July 2014. M and A samples from 2014 are marked with gold circles and red squares, respectively. The 2016 spring region is highlighted with a dashed gold line, and the location of the spring is marked with a black triangle. Samples from 2017 were collected from within yellow dashed area. The inset line map shows the approximate location of Borup Fiord Pass (red dot) on Ellesmere Island, NU, Canada. (Satellite image courtesy of Maxar Technologies; inset line map courtesy of Wikimedia Commons and the CC 3.0 license.)

Microbial sulfur oxidation and reduction, which spans an eight-electron transfer between the most highly oxidized (sulfate) and reduced (hydrogen sulfide) states, is metabolically important in a multitude of diverse extreme environments, such as hydrothermal vents and deep subsurface sediments (14), microbial mats (15–17), sea ice (18), glacial environments (19, 20), and Arctic hypersaline springs (3, 21). Two predominant forms of sulfur-utilizing microorganisms exist: sulfur-oxidizing microorganisms (SOMs), which use reduced compounds (e.g., $H_2S$ and $S^0$) as electron donors, and sulfate-reducing microorganisms (SRMs), which use oxidized forms of sulfur (e.g., $SO_4^{2-}$ and $SO_3^{2-}$) as electron acceptors. SOMs are metabolically and phylogenetically diverse (14, 22, 23) and can often fix carbon dioxide ($CO_2$) using a variety of electron donors (24). Likewise, SRMs utilize a range of electron donors, including organic carbon, inorganic carbon, hydrogen, and metallic iron (25). They can also use heavy metals—for example, uranium, where soluble U(VI) is converted to insoluble U(IV) under anoxic conditions—as electron acceptors (26, 27). SRMs are particularly vital in sulfate-rich marine environments, where they contribute as much as 50% of total global organic carbon oxidation (28, 29).

Previous research has been conducted on microbial sulfur cycling in low-temperature polar environments in both Antarctica (30, 31) and the Arctic (3, 19), including BFP (9, 32, 33). However, the organisms participating in sulfur cycling and the metabolic pathways that they use in these low-temperature environments remain understudied. It is important that we better classify these systems as they are key to our understanding of how life can adapt to potentially adverse conditions, such as low temperatures, highly sulfidic conditions, and low carbon and nutrient levels. Furthermore, these microbial adaptations inform us about where to search for potential extraterrestrial life on other planetary bodies. One example where this research is applicable is Europa, where we know that low-temperature, sulfur-rich conditions exist (34–36).

We collected samples over multiple years (2014, 2016, and 2017; Fig. 2) and from various sample types (e.g., aufeis, melt pools, spring fluids, and surface mineral precipitates; see Fig. S2 in the supplemental material) for metagenomic whole-genome sequencing (WGS). The sequencing data were assembled and binned into metagenome-assembled genomes (MAGs) and sample metagenomes (metagenome assemblies of the site samples themselves). These assemblies were used to identify metabolic pathways and their completeness across the BFP samples. Analysis of the MAGs and metagenomes revealed the presence of sulfur oxidation genes (namely, those involved in thiosulfate oxidation) across multiple phyla, including those related to organisms where sulfur oxidation is not predicted to be metabolically viable. This may indicate a form of functional redundancy present in the BFP system where organisms from other phyla can take advantage of the abundance of reduced sulfur for metabolic processes.

## RESULTS

**Assembly and identification of metagenome-assembled genomes and sample metagenomes.** High-throughput sequencing of total environmental genomic DNA (gDNA) extracted from nine biomass samples collected from the BFP spring/aufeis/mineral precipitates produced paired-end reads that were trimmed and assembled *de novo*. Total assembly lengths of sample metagenomes ranged from 638 Mbp (site 14C) to 10.1 Mbp (site M2). The coassembly of all sites was 1.16 Gbp in length, made up of 477,394 contigs, the largest being 406,785 bp long and the average being 2,440 bp long. A summary of statistics (total base pairs, number of contigs, maximum contig length, average contig length, and $N_{50}$) for individual site assemblies and the coassembly is available in Table 2. After assembly, a total of 166 bins were identified using CONCOCT (37) and manual refinement within Anvi'o (38). Of these, 31 were classified as medium-quality MAGs (defined as bins with more than 50% estimated genome completeness and less than 10% redundancy [often referred to as contamination]) by MIMAG (Minimum Information about Metagenome-Assembled Genomes; 39) stan-

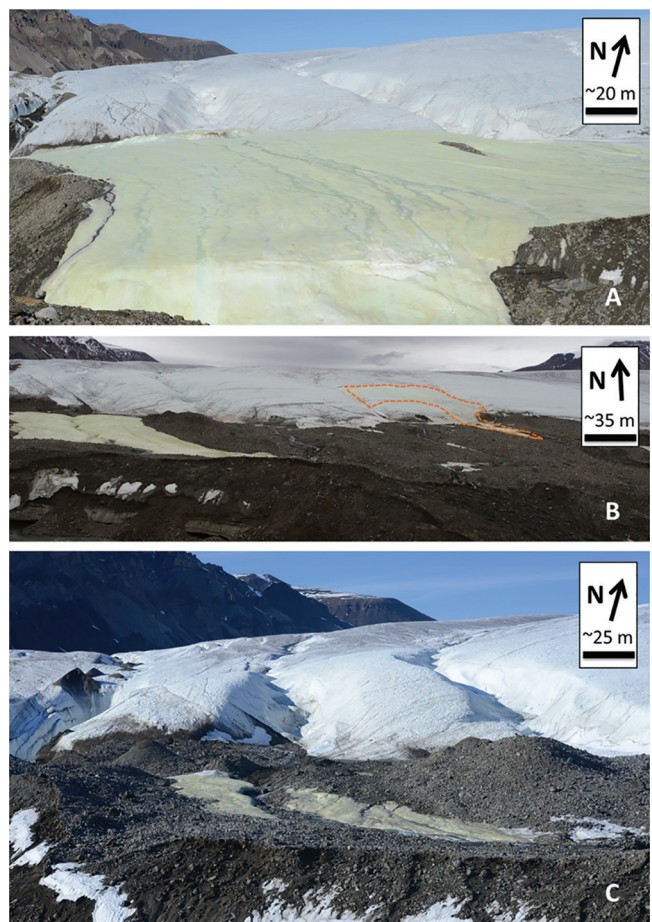

**FIG 2** BFP sampling locations. (A) The "Sulfidic Aufeis" (from Trivedi et al.; 13) during 2014. Note that two sample locations are to the southwest of the area shown in this picture; their approximate locations can be seen in Fig. 1. (B) The sulfidic aufeis on the left of the picture with the 2016 spring a few hundred meters to the east (indicated by the orange dashed line; also see Fig. S1). (C) The same location and focus of the sampling in 2017.

dards. Of the 31 qualified MAGs, nine were over 90% complete with less than 10% redundancy. Further summary statistics for MAGs are available in Table 1. The levels of MAG read coverage of the different BFP sites differed (Fig. 3). For example, MAG 31 (most closely related to the genus *Flavobacterium*) had abundant read contributions from samples A6 (aufeis), AS3b, AS4e, and AS6b (all mineral precipitate samples) and moderate contributions from A4b and M4b (aufeis and melt pool samples, respectively). However, MAG 31 was also the least complete of the reported MAGs according to its genome completeness percentage.

All MAGs were taxonomically classified as *Bacteria*, the majority of which (17 of 31) were within the phylum *Proteobacteria* (Table 1). Of these, eight belonged to the *Gammaproteobacteria* (MAGs 2, 4, 11, 15, 16, 23, 27, and 29), five to the *Betaproteobacteria* (MAGs 5, 6, 14, 19, and 20), four to the *Alphaproteobacteria* (MAGs 3, 10, 24, and 25), two to the *Desulfobacterota* (MAGs 7 and 12), one to *Desulfuromonadota* (MAG 21), and two to the *Campylobacteria* (MAGs 1 and 9). Each of the proteobacterial classes had at least one MAG considered highly (>90%) complete with low (<10%) redundancy (Table 1).

**Non-sulfur-associated metabolic potential identified across BFP.** Metabolic pathways queried for this study are presented in a modified heat map (Fig. 4), while the entire pathway map can be found in Fig. S3 in the supplemental material. The pathway heat map shown in Fig. 4 details the completeness of each of these pathways based on

**TABLE 1** Metadata for all reported MAGs[a]

| MAG | Putative taxonomy | Length (bp) | Total no. of contigs | $N_{50}$ (bp) | % GC | % completion | % redundancy |
|---|---|---|---|---|---|---|---|
| 1 | p_Epsilonbacterota; g_Sulfurimonas | 2,064,274 | 14 | 214,847 | 38.19 | 99.59 | 0 |
| 2 | c_Gammaproteobacteria; g_Shewanella | 4,979,965 | 67 | 119,878 | 41.35 | 99.45 | 1.12 |
| 3 | c_Alchaproteobacteria; o_Rhizobiales | 7,775,429 | 338 | 70,375 | 63.24 | 99.15 | 3.72 |
| 4 | c_Gammaproteobacteria; g_Cellvibrio | 3,581,784 | 67 | 99,909 | 43.59 | 98.28 | 2.46 |
| 5 | c_Gammaproteobacteria; g_Limnobacter | 3,445,427 | 282 | 19,477 | 52.80 | 97.39 | 1.13 |
| 6 | c_Alphaproteobacteria; f_Sphingomonadaceae | 3,009,949 | 326 | 11,575 | 58.93 | 93.54 | 1.92 |
| 7 | c_Desulfobacterota; g_Desulfocapsa | 2,805,299 | 284 | 13,203 | 51.33 | 92.84 | 2.28 |
| 8 | p_Actinobacteria; c_Coriobacteriia | 2,150,593 | 263 | 13,565 | 60.87 | 92.5 | 3.33 |
| 9 | p_Epsilonbacterota; g_Sulfurovum | 1,833,503 | 136 | 22,712 | 38.75 | 92.21 | 1.84 |
| 10 | c_Alphaproteobacteria; g_Loktanella | 3,281,266 | 483 | 7,930 | 60.60 | 89.02 | 2.3 |
| 11 | c_Gammaproteobacteria; g_Arenimonas | 2,721,939 | 304 | 11,951 | 68.73 | 87.86 | 1.12 |
| 12 | p_Desulfobacterota; g_Desulfocapsa | 2,954,298 | 447 | 7,903 | 50.14 | 84.94 | 2.08 |
| 13 | p_Bacteroidetes; g_Algoriphagus | 4,253,479 | 791 | 5,920 | 41.69 | 80.28 | 9.29 |
| 14 | c_Gammaproteobacteria; g_Herminiimonas | 2,209,716 | 410 | 5,958 | 51.10 | 79.43 | 3.13 |
| 15 | c_Gammaproteobacteria; g_Marinobacter | 3,818,186 | 748 | 5,456 | 54.16 | 78.56 | 5.04 |
| 16 | c_Gammaproteobacteria; g_Pseudomonas | 6,369,279 | 177 | 60,989 | 58.13 | 77.54 | 0.88 |
| 17 | c_Chloroflexi; c_Chloroflexia | 4,795,723 | 834 | 6,374 | 49.05 | 74.65 | 3.67 |
| 18 | p_Bacteroidetes; f_Sphingobacteriaceae | 3,604,420 | 465 | 10,926 | 33.46 | 73.33 | 5.44 |
| 19 | c_Gammaproteobacteria; g_Thiobacillus | 2,632,478 | 497 | 5,833 | 61.30 | 72.78 | 1.57 |
| 20 | c_Gammaproteobacteria; g_Rhodoferax | 2,881,094 | 579 | 5,077 | 59.36 | 72.38 | 3.83 |
| 21 | p_Desulfuromonadota; o_Desulfuromonadales | 2,948,378 | 557 | 5,659 | 51.99 | 69.53 | 0.65 |
| 22 | p Elusimicrobia; o_Elusimicrobiales | 2,483,169 | 437 | 6,174 | 58.20 | 69.41 | 2.35 |
| 23 | c_Gammaproteobacteria; g_Thiomicrospira | 2,196,521 | 426 | 5,432 | 42.35 | 68.46 | 7.66 |
| 24 | c_Alphaproteobacteria; g_Sandarakinorhabdus | 2,926,576 | 607 | 4,990 | 63.80 | 67.43 | 1.12 |
| 25 | c_Alphaproteobacteria; g_Sandarakinorhabdus | 2,352,194 | 538 | 4,519 | 64.93 | 62.24 | 4.89 |
| 26 | p_Planctomycetes; c_Planctomycetia | 2,408,829 | 457 | 5,553 | 45.58 | 61.92 | 2.43 |
| 27 | c_Gammaproteobacteria; g_Hydrogenophaga | 2,910,411 | 640 | 4,770 | 62.31 | 61.79 | 0.44 |
| 28 | c_Deinococci; f_Deinococcaceae | 3,481,023 | 613 | 6,273 | 60.43 | 59.34 | 2.97 |
| 29 | c_Gammaproteobacteria; f_Burkholderiaceae | 1,596,490 | 423 | 3,649 | 52.68 | 55.17 | 8.31 |
| 30 | p_Firmicutes; c_Bacilli | 974,526 | 65 | 28,355 | 33.20 | 51.75 | 4.55 |
| 31 | p_Bacteroidetes; g_Flavobacterium | 1,930,691 | 418 | 4,726 | 34.69 | 51.45 | 2.36 |

[a]All MAGs shown represent >50% completion with <10% redundancy (contamination) via CheckM. Taxonomy was determined by consensus between GTDB and MiGA.

the presence or absence of key enzymes. Carbon fixation potential was inferred by evaluating the completeness of five pathways, including the Calvin-Benson-Bassham (CBB), reverse tricarboxylic acid (rTCA), Wood-Ljundahl, 3-hydroxypropionate, and 4-hydroxybutyrate/3-hydroxypropionate pathways. The CBB cycle was complete or nearly complete in four MAGs (10, 27, 19, and 23) and in all but one sample metagenome (M2). The rTCA cycle was complete in only two MAGs (1 and 9), both of which were taxonomically classified within the *Campylobacteria*. Additionally, the rTCA cycle was complete in 5 of 9 sample metagenomes (A6, A4b, M2, Spring 2016, and AS3b). The Wood-Ljungdahl pathway was approximately 70% complete (Fig. 4) in MAGs 8 and 7 and in sample metagenomes M2 and AS3b. Finally, the 3-hydroxypropionate and 4-hydroxybutyrate/3-hydroxypropionate cycles were the least complete (<50%) across MAGs but were widely distributed across all sample metagenomes (Fig. 4).

Metabolic pathways involved in denitrification were also considered, as the reduction of oxidized nitrogen species can be utilized in sulfur oxidation reactions. Fully complete dissimilatory nitrate reduction (to nitrite) pathways were present in a number of MAGs (12, 9, 1, 27, 2, 4, and 15) as well as in all but AS4e in the sample metagenomes. Additionally, three MAGs (12, 27, and 4) contained genes for a complete dissimilatory nitrate reduction to ammonium (DNRA; $NO_3^-$ to $NH_4^+$) pathway. Two MAGs (7 and 12, both of which were classified within the genus *Desulfocapsa*) also contained genes for the complete pathway for nitrogen fixation. Many of the other MAGs as well as the sample metagenomes showed complete additional pathways such as nitrite, nitric oxide, and nitrous oxide reduction (Fig. 4).

Multiple genes associated with pathways responsible for aerobic respiration were present as part of the MAGs and sample metagenomes. The mostly likely pathways for

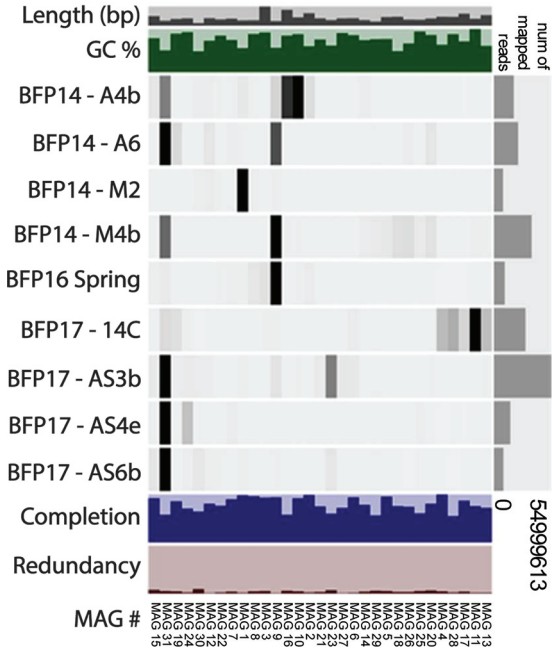

**FIG 3** Anvi'o diagram relating sites and MAGs. This diagram shows all MAGs with greater than 50% completion and less than 10% redundancy. Higher read recruitment to a given MAG by a site sample is indicated by darker bars, while lower recruitment is indicated by lighter bars or no bars. A bar plot representing the total number of mapped reads for a given site sample is on the right side. Also shown are bar representations of total length (in base pairs) and (percent) GC content of sites as part of the coassembly. Completion and redundancy bar plots for MAGs echo those statistics presented in Table 1.

oxidative phosphorylation within the BFP samples were via F-type ATPase, ubiquinol-cytochrome *c* reductase, cytochrome *c* oxidase (*cbb3* type), and cytochrome *bd* complex (Fig. 4).

**Sulfur oxidation is abundant across BFP metagenomes and MAGs.** Of greatest interest for this study were sulfur cycling-associated genes (including genes involved in both the oxidation and reduction of sulfur species). MAGs and sample metagenomes were queried for metabolic pathways and for well-known genes that encode enzymes involved in sulfur-based metabolism as follows: sulfide oxidation (e.g., *fcc*, *sqr*), sulfur oxidation (*sdo*; sulfur dioxygenase), sulfite dehydrogenase (*sorB*), and thiosulfate oxidation (*sox* and *tsdA*). Full pathways for sulfide oxidation were found within 11 of the 31 MAGs and all of the sample metagenomes (Fig. 4). All but three MAGs (31 [*Flavobacterium*], 9 [*Sulfurovum*], and 1 [*Sulfurimonas*]) were part of the *Proteobacteria*. The remaining MAGs were classified within both the alphaproteobacterial and gammaproteobacterial classes. Ten MAGs and nine sample metagenomes contained the gene that encodes sulfur dioxygenase (*sdo*), which is often responsible for the oxidation of elemental sulfur. Additionally, 15 of 31 MAGs and all sample metagenomes had genes necessary to encode sulfite dehydrogenase (*sorB*).

The Sox enzyme complex is widely studied for its ability to facilitate the complete oxidation of thiosulfate ($S_2O_3^{2-}$) to sulfate ($SO_4^{2-}$). The presence of the thiosulfate pathway at BFP was determined from the presence of genes *soxA*, *soxB*, *soxC*, *soxX*, *soxY*, and *soxZ*. Pathway completion was measured as a fraction of the levels of these genes. Complete Sox pathways were present in MAGs 9, 1, 10, and 15, with partially complete pathways found in MAGs 14, 27, 5, 29, 20, 19, and 23. Additionally, the complete thiosulfate oxidation pathway was present in all nine of the sample metagenomes. All but two of these MAGs (MAGs 9 and 1, *Campylobacteria*) were classified within the *Proteobacteria*, the majority being *Gammaproteobacteria* (MAGs 14, 27, 5, 29, 20, 19, 15, and 23) and the remaining being *Alphaproteobacteria* (MAG 10). Additionally, an alternative thiosulfate oxidation pathway, represented by the *tsdA* gene, was found to

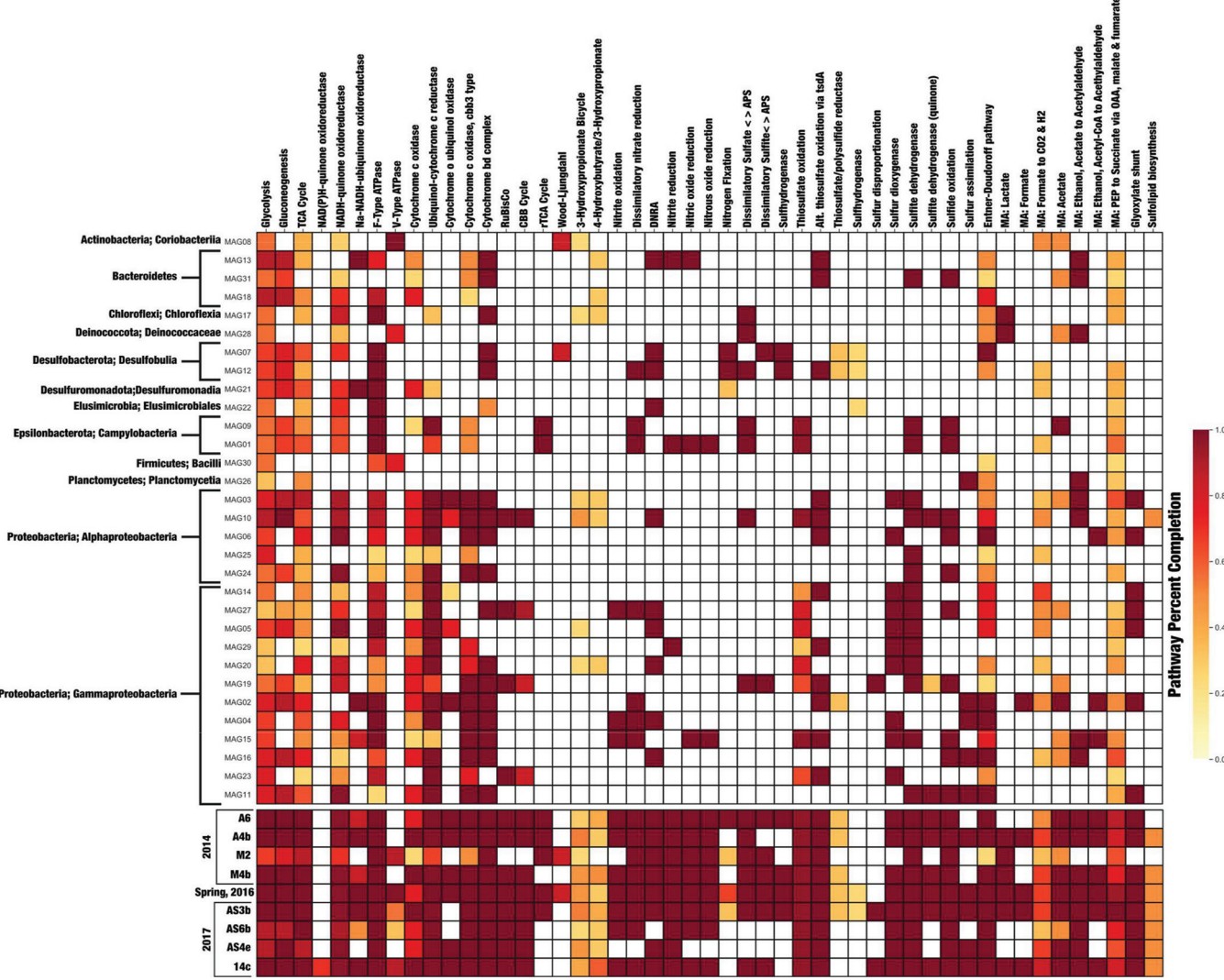

**FIG 4** KEGG Decoder heat map based on Kofamscan results. The heat map represents metabolic pathway completeness based on the presence or absence of genes as determined by KEGG Decoder. Dark red (the color scale is shown at right) represents a complete or highly complete pathway, while white represents locations where a pathway is absent or highly incomplete. The data represent a subset of results as described in the manuscript; the full set of pathways is shown in Fig. S3. Additionally, sample metagenomes are below MAGs in chronological order. MAGs are ordered based on taxonomic class. OAA, oxaloacetic acid; PEP, phosphoenolpyruvate.

be present in 12 of 31 MAGs and all of the sample metagenomes. Six of these MAGs (13, 31, 12, 3, 6, and 2) contained the *tsdA* gene but did not contain any *sox* genes.

Also queried was reversible dissimilatory sulfate reduction, which is shown in Fig. 4 separated into "Dissimilatory sulfate < > APS" (adenosine 5′-phosphosulfate; *sat* and *apr*) and "Dissimilatory sulfite < > APS" (*dsr*). These cover the complete reduction of sulfate to sulfide as *sat* and *apr* genes encode enzymes responsible for the reduction of sulfate to sulfite (via the intermediate APS) and then *dsr* encodes enzymes that reduce sulfite the remainder of the way to sulfide (40). Multiple MAGs (17, 28, 12, 9, 1, 10, and 19) as well as all but two (AS6b and AS4e) sample metagenomes had both *sat* and *apr* genes. Those MAGs taxonomically corresponded to *Chloroflexia* (class), *Deinococcus*, *Desulfocapsa*, *Sulfurovum*, *Sulfurimonas*, *Loktanella*, and *Thiobacillus*, respectively. Conversely, only two MAGs (7 and 19) and all but three sample metagenomes (A4b, AS6b, and AS4e) were found to have the *dsr* genes needed for the reduction of sulfite to sulfide. Sulfur disproportionation was identified based on the presence of the *sor* (sulfur oxygenase/reductase) gene. This gene was found only in MAG 19, which was

classified as *Thiobacillus*. The *sor* gene was also present in two of the sample metagenomes (AS3b and 14c).

## DISCUSSION

BFP is a polar, low-temperature, glacial ecosystem in the Canadian High Arctic dominated by sulfur-rich water and ice along with precipitated minerals. The bioenergetic capability of microbes living in a low temperature, sulfur-rich sediment mound formed on the glacier surface at BFP was previously identified through metagenomic sequencing (9). Here, we greatly enhanced our understanding of low-temperature microbial metabolisms by expanding the number of sites and sample types beyond just surface precipitates. We examined aufeis samples, melt pool samples, and fluid from the active spring discharge. Through the use of metagenomic sequencing, we have identified microbial community members at BFP that have an abundance of genes encoding enzymes involved in the oxidation of reduced sulfur species, including taxa known to perform this process (e.g., *Campylobacteria*—*Sulfurimonas* and *Sulfurovum*) as well as others not previously known to oxidize sulfur in these environments (e.g., *Limnobacter* sp.).

In addition to sulfur cycling, we also looked at core metabolic processes, including carbon and nitrogen metabolisms. Low levels of nitrate were previously reported from samples collected at BFP (0.0030 mM in 2009 [9] and 0.0908 mM in 2014 [10]) as part of spring and melt pool fluids, respectively—possibly due to microbial nitrate reduction. On the basis of metabolic pathway analysis results, we infer that some organisms from BFP samples may be capable of nitrate/nitrite reduction. Dissimilatory nitrate reduction to nitrite using genes *nap* (periplasmic nitrate reductase) and *nar* (nitrate reductase) was found in multiple MAGs (1, 2, 4, 9, 14, 26, and 31) and sample metagenomes (all but AS4e). Further reduction of nitrite to ammonium encoded by *nir* (nitrite reductase) and *nrf* was present in only three MAGs (12, 27, and 4) and in all but one sample metagenome (AS6b), meaning that only three of the putative organisms were capable of full reduction of nitrate to ammonium. Note also that nitrate reduction was found in both MAG 1 (*Sulfurimonas*) and MAG 9 (*Sulfurovum*), suggesting that SOMs such as *Sulfurimonas* and *Sulfurovum* found in higher relative abundance (13) at some sample sites (e.g., M2 and M4b, respectively) are capable of oxidizing reduced sulfur compounds and utilizing nitrate as a terminal electron acceptor.

Oxygen is certainly present at air-surface interfaces across BFP, and the majority of organisms present most likely use oxygen as an electron acceptor. However, metagenomic evidence also supports the idea of the presence of denitrification as a viable mechanism of low-temperature respiration. Additionally, two MAGs (7 and 12), classified as belonging to the genus *Desulfocapsa*, contained genes required for nitrogen fixation. Even at relatively low abundances (13), they may provide some form of community control by moderating the addition of organic N into the system. Genes related to aerobic respiration were also present in abundance across BFP MAGs and sample metagenomes. The genes that encode F-type ATPase were found to be abundant across the majority of MAGs and throughout all of the sample metagenomes. The majority of other aerobic respiration pathways were mostly complete within MAGs related to the *Proteobacteria*, including those that use ubiquinol-cytochrome *c* reductase, cytochrome *c* oxidase (*cbb3* type), and cytochrome *bd* complex enzymes. These pathways were mostly complete across all sample metagenomes as well. Cytochrome *c* oxidases have previously been shown to operate under conditions of low oxygen tension (41) and could be useful in hypoxic zones near surface sediments; however, this has not yet been established for the *cbb3* cytochrome oxidases such as those identified here.

Understanding the utilization of carbon in microbial growth dynamics at the BFP site is vital to determining how carbon might enter the system. Organic carbon is available but is low in concentration at BFP, ranging from 0.07% to 0.20% C (10), and can also potentially be sourced from nearby shale units (7). Previous research by Wright et al. (9) indicated that the most likely form of energy production in the system is via aerobic

oxidation of $S^0$ and that chemolithoautotrophy is the main form of primary production at the site. Similarly to the results reported by Wright et al., our metagenomic data support the finding that $CO_2$ fixation rather than direct oxidation of organic carbon is likely one of the most common metabolisms present at BFP. We identified genes in MAGs for five major carbon fixation pathways: the Calvin-Benson-Bassham (CBB), reverse tricarboxylic acid (rTCA), Wood-Ljundahl, 3-hydroxypropionate, and 4-hydroxy-butyrate/3-hydroxypropionate pathways. One well-described phylum containing organisms capable of autotrophy (e.g., carbon fixation) and sulfur oxidation is the *Campylobacteria* phylum (42). Additionally, we identified four MAGs (MAG 10, MAG 27, MAG 19, and MAG 23) within the *Proteobacteria* (corresponding to *Loktanella*, *Hydrogenophaga*, *Thiobacillus*, and *Thiomicrospira*, respectively) that contained a complete or nearly complete CBB cycle. This includes the gene that encodes RuBisCO (ribulose 1,5-bisphosphate carboxylase/oxygenase), a key enzyme involved in carbon fixation. Another possible fixation pathway, the rTCA cycle pathway, was found in MAGs within the members of *Campylobacteria*, specifically, *Sulfurimonas* and *Sulfurovum* (MAGs 1 and 9, respectively), as well as in five of our sample metagenomes (A6, A4b, M2, Spring 2016, and AS3b). *Sulfurimonas* and *Sulfurovum* were previously identified by 16S rRNA gene sequencing at BFP and were assumed to play a key role in sulfur cycling (13). The identification of the rTCA cycle in organisms found in abundance within this study and previously via 16S rRNA gene sequencing suggests that this pathway, coupled to sulfur oxidation, represents a likely mechanism of carbon incorporation at BFP (14, 43).

Three less-common carbon incorporation pathways at BFP were the Wood-Ljungdahl pathway and the 3-hydroxypropionate and 4-hydroxybutyrate/3-hydroxypropionate cycles. The Wood-Ljunghdahl pathway was only partially identified in two MAGs (8 and 7) that are taxonomically related to class *Coriobacteriia* and genus *Desulfocapsa*. This agrees with a previous study, where it was reported that *Desulfocapsa sulfexigens* was able to thrive on $CO_2$ as its sole carbon source via the reverse acetyl coenzyme A (acetyl-CoA) (Wood-Ljungdahl) pathway (44). MAG 12 was also classified as corresponding to *Desulfocapsa* but was not as complete as MAG 7 (84.9% versus 92.8%, respectively; Table 1). This small difference in genome completion might explain why genes associated with the Wood-Ljungdahl pathway were found in one (MAG 7) but not the other (MAG 12). Additionally, the Wood-Ljungdahl pathway was found to be only partially complete in sample metagenomes M2 and AS3b, between which M2 was shown to have a higher abundance of *Desulfocapsa* in the data reported in 2018. The 3-hydroxypropionate and 4-hydroxybutyrate/3-hydroxypropionate cycles were only partially complete across a number of MAGs and sample metagenomes (Fig. 4). While a number of different carbon fixation pathways appear to be present and possibly utilized by BFP microorganisms, the CBB and rTCA cycles are the most complete and therefore appear to represent the preferred methods of incorporation of carbon at BFP.

Core to our work was the identification of genes and pathways related to sulfur cycling. The research presented here identified sulfur cycling genes in MAGs and sample metagenomes across samples from BFP and across multiple different taxonomic lineages. Previous studies of both spring fluid and mineral deposits at BFP produced reports of the presence of a number of the same organisms as were identified by our MAGs, including *Marinobacter*; *Loktanella* (7, 32); and the sulfur oxidizers *Sulfurimonas*, *Sulfurovum*, *Sulfuricurvum*, and *Thiobacillus* (7, 9, 32, 45). One organism showing full completion of multiple sulfur cycling pathways was *Loktanella* (MAG 10). This MAG shows complete pathways for sulfide oxidation, sulfite oxidation, two forms of thiosulfate oxidation, and (reversible) dissimilatory sulfate reduction. *Loktanella* spp. were also previously identified at BFP in 16S rRNA gene clone libraries (32) as well as in other low-temperature environments throughout the Canadian Arctic (3, 46, 47). *Loktanella* was previously reported to contain *soxB* on the basis of samples obtained at Gypsum Hill, a perennial Arctic spring on Axel Heiberg Island (21). The study by Perreault et al. (21) also revealed that the *Loktanella* sp. was part of a consortium that also contained a *Marinobacter* sp., similarly to previous findings at BFP (32). The genome for *Marino-

*bacter* (MAG 15) analyzed in this study was also found to have a number of complete sulfur cycling pathways, including pathways associated with sulfide oxidation, sulfur oxidation (via sulfur dioxygenase), both forms of thiosulfate oxidation, and sulfite oxidation (via sulfite dehydrogenase). MAG 5, putatively identified as representing the genus *Limnobacter*, contains an almost complete thiosulfate oxidation pathway (Fig. 4) based on genes associated with the Sox complex of enzymes. Chen et al. (48) previously reported the full *sox* pathway as part of a *Limnobacter* sp. genome in an anaerobic methane-oxidizing microbial community, while other studies have reported its ability to oxidize thiosulfate to sulfate (49, 50).

We identified several new organisms not previously reported by 16S rRNA gene sequencing analyses, including the *Betaproteobacterial* genera *Herminiimonas* and *Rhodoferax*. Despite its identification in a 120,000-year-old Greenland ice core (51), *Herminiimonas* is rarely identified in polar environments. Moreover, data on the ability of *Herminiimonas* to oxidize sulfur are equally sparse. In a recent study, Koh et al. (41) found that *Herminiimonas arsenitoxidans* (which has been found to oxidize arsenite) was able to oxidize sulfur when grown on Trypticase soy agar. Our *Herminiimonas* (MAG 14) contained a partially (<50%) complete thiosulfate oxidation pathway, a complete alternative thiosulfate oxidation pathway, and genes that encode sulfur dioxygenase (*sdo*) and sulfite dehydrogenase (*sorB*), supporting the *in vitro* work by Koh et al. suggesting that *Herminiimonas* has the ability to oxidize sulfur.

Another common polar microorganism, *Rhodoferax* (also known as purple nonsulfur bacteria) (52, 53), was identified in our work. *Rhodoferax* bacteria are facultatively photoheterotrophic, and some are capable of photoautotrophy when sulfide is used as an electron donor (54). We identified one *Rhodoferax* genome (MAG 20) containing a nearly complete (approximately 70%) thiosulfate oxidation pathway as well as complete pathways for sulfur oxidation and sulfite oxidation. While not present in any great amount, it is possible that *Rhodoferax* sp. contribute to the total amount of oxidized sulfur species in the system.

The SoxCD enzyme complex has been shown to be responsible for the oxidation of sulfur to thiosulfate, and in organisms that lack this complex, the sulfur is either stored inside the cell or excreted (55). The lack of the *soxC* gene in BFP MAG 19 of the known sulfur oxidizer *Thiobacillus* is an interesting finding. In fact, this represents one possible biological explanation for the abundance of $S^0$ precipitated across the surface of BFP. However, it seems more likely that only the *soxC* gene is missing, as the genome for MAG 19 is only 72% complete (Table 1). Abiotic sulfur oxidation is predicted to occur at this site based on thermodynamics (9); however, at such low temperatures, this process may be kinetically slow without biological catalysis (and this may explain the large abundance and year-to-year persistence of elemental sulfur at the site). Regardless of the accumulation of $S^0$ in the BFP system, some organisms may employ other pathways such as the use of reverse dissimilatory sulfite reduction (Dsr, encoded by *dsrAB*) to oxidize accumulations of $S^0$ (56), as may be the case with MAGs 7 (*Desulfocapsa*) and 19 (*Thiobacillus*), which both contain genes for this pathway. We also investigated which electron donors might support sulfur oxidation at BFP. Specifically, the hydrogen:quinone oxidoreductase pathway was found to be present in MAGs 1, 9, and 15 (Fig. 4), which are classified to known SOMs (*Sulfurimonas*, *Sulfurovum*, and *Thiobacillus*, respectively) in the system. The discovery of the hydrogen:quinone oxidoreductase pathway, a metabolic pathway that uses molecular hydrogen as an electron donor for the reduction of quinone, is intriguing in the face of (unpublished) data collected in 2014, where 29 nM $H_2$ was measured in one of the melt pools (site M4; 13). This indicates that molecular hydrogen oxidation operating through this pathway could act as an alternative to the oxidation of sulfur.

Microbially mediated sulfur oxidation is undoubtedly a key component of the ecology of the BFP spring site, and such a large amount of sulfate (ranging from 0.06 to 14.91 mM for these samples; 13) would also support a community of anaerobic microorganisms capable of dissimilatory sulfate reduction. Of the 31 MAGs identified, 7 (MAGs 17, 28, 12, 9, 1, 10, and 19; Fig. 4) were found to contain dissimilatory sulfate

reduction (*sat* and *apr*) genes, representing a key first step in the complete reduction of sulfate to sulfide. However, the next step, genes involved in dissimilatory sulfite reduction (encoded by *dsrAB*) were found to be present in only two MAGs (7 and 19), which correspond to *Desulfocapsa* and *Thiobacillus*, respectively. The members of the genus *Desulfocapsa* are known sulfur disproportionators (44, 57), and were previously identified at BFP via 16S rRNA gene sequencing (13); however, the *sor* gene queried for sulfur disproportionation was found only in the *Thiobacillus* sp. (MAG 19). Interestingly, MAG 19 is the only MAG that appears capable of complete sulfate reduction via the dissimilatory sulfate and sulfite reduction pathways. *Thiobacillus* spp. are traditionally known as sulfur oxidizing microorganisms and have been found in polar environments (19, 58, 59); however, it has also been suggested that certain *Thiobacillus* species carry out reverse dissimilatory sulfite reduction (rDsr, or reverse Dsr) as an alternative means to oxidize reduced sulfur compounds (19, 60). This could represent an alternative pathway that is present in the BFP system; however, we can only speculate without transcriptomic evidence. Note that MAG 19 appears to lack the *soxC* and *soxX* genes, though the alternative thiosulfate oxidation pathway is present. Furthermore, MAG 19 is approximately 72% complete (Table 1), which may indicate that its full metabolic potential is not represented.

Elemental sulfur is an obvious visual component of the Borup ecosphere. Unsurprisingly, the members of a broad group of organisms have adapted and even come to coexist to take advantage of this abundant element at BFP. Functional redundancy is common in other cold organic carbon-limited environments such as deep subseafloor aquifers (61). Louca et al. (62) recently identified functional redundancy as a commonplace process in environmental systems and have speculated that the degree of functional redundancy in an ecosystem is largely determined by the type of environment, in contrast to the belief that species should inhabit distinct microbial niches in a manner independent of environment. Moreover, the authors also called into question the previously reported idea that functional redundancy in an ecosystem might imply a form of "neutral coexistence" of competing microorganisms (63). The broad taxonomic distribution of microorganisms capable of sulfur oxidation at BFP seems to suggest that a certain level of community metabolic redundancy rather than a core group of specialists exists. Previously, we suggested that a core microbial community existed at BFP and yet was not involved in sulfur cycling (13). Instead, our current metagenomic data indicate that members of this core community identified by 16S rRNA gene sequencing may indeed be capable of sulfur oxidation.

Louca et al. (62) suggested that microorganisms in many environments may not be equally abundant even when they have similar metabolic capabilities as there are always differences in enzyme efficiencies and growth kinetics. Their hypothesis of functional redundancy is also supported within the BFP system, where "blooms" of known SOMs occur over short time periods within transient pools of liquid water, followed by a return to the basal microbial community (13). Conditionally rare taxa, such as those identified as part of the core community at BFP (i.e., corresponding to the low level of representation of microbiota in any environment that can become dominant upon environmental condition change), might represent a pool of functional redundancy, allowing sulfur oxidation to occur even when conditions become untenable for known SOMs. There exist numerous extreme pressures at BFP, including extreme cold, low organic carbon, high summer insolation followed by months of darkness, and high concentrations of multiple sulfur species (e.g., $H_2S$, $HS^-$, $S^0$, $S_2O_3^{2-}$, and $SO_4^{2-}$; 9). All of these factors place pressure on the continued survival of a microbial community. Still, blooms of life obviously occur at Borup, and these blooms might be facilitated by the input of new microbiota via Aeolian transportation from pool to pool in storm events or even from distant lands, whereby snow storms contain significant amounts of microbiota and genetic potential (64). When there is a large influx of reduced sulfur compounds to the surface from beneath the ice, some phyla, like the occasionally (conditionally) rare *Campylobacteria* (13), may be preferentially able to use these compounds (much like the results seen in investigations of deep sea vent fluids;

**TABLE 2** Sample metagenome and coassembly metadata[a]

| Sample | Source | Total length (bp) | Total no. of contigs | Maximum contig length (bp) | Avg contig length (bp) | $N_{50}$ (bp) |
|---|---|---|---|---|---|---|
| BFP14_A4b | Aufeis | 85,484,280 | 30,017 | 380,944 | 2,848 | 3,571 |
| BFP14_A6 | Aufeis | 82,535,183 | 33,359 | 128,624 | 2,474 | 2,871 |
| BFP14_M2 | Melt pool | 10,144,305 | 4,841 | 418,178 | 2,095 | 1,977 |
| BFP14_M4b | Melt pool | 158,556,086 | 71,593 | 111,481 | 2,215 | 2,359 |
| BFP16_Spring | Spring fluid | 60,894,340 | 18,996 | 206,795 | 3,206 | 4,748 |
| BFP17_14C | Aufeis surface | 638,405,366 | 272,035 | 297,076 | 2,347 | 2,611 |
| BFP17_AS3b | Aufeis surface | 197,768,313 | 78,690 | 78,857 | 2,513 | 3,028 |
| BFP17_AS4e | Aufeis surface | 89,868,307 | 27,395 | 44,699 | 3,280 | 4,468 |
| BFP17_AS6b | Aufeis surface | 100,119,525 | 34,932 | 60,991 | 2,866 | 4,186 |
| | | | | | | |
| Coassembly | | 1,164,607,597 | 477,394 | 406,785 | 2,440 | 2,778 |

[a]These data represent raw values from assembly performed through Megahit prior to read mapping with Bowtie2 and MAG classification via Anvi'o.

65), leading to an increase in their relative abundance until they resettle to previous levels. Undoubtedly, sulfur oxidation is the predominant and preferred form of metabolism in these sulfur-rich surface sites at BFP.

In low-temperature environments, functional redundancy of key metabolisms is a necessary strategy for the coexistence and survival of multiple microbial lineages. The results presented here reflect the potential for the presence of diverse metabolisms at BFP and the identification of microorganisms not previously thought to participate in low-temperature sulfur cycling. Future work will define how microorganisms are able to survive and thrive in the BFP system via metatranscriptomic sequencing by differentiating between metabolic potential and metabolic actuality across sample types at BFP. Still, this glimpse at the metabolic potential and taxonomic diversity present at such a unique low-temperature environment at Borup Fiord Pass shows not only that even well-described and well-studied pathways such as the sulfur oxidation pathway are functionally redundant but also that they may be utilized by more microorganisms across the bacterial domain than expected.

## MATERIALS AND METHODS

**Sample collection.** BFP sample material was collected during multiple days between 21 June and 2 July 2014, 4 July 2016, and 7 July 2017. Samples included (i) sedimented sulfur-like cryoconite material and fluid found in glacial surface melt pools (labeled M2 and M4b), (ii) filtered fluid from thawed aufeis (labeled A4b and A6), (iii) aufeis surface precipitate samples (material scraped from the top of the aufeis) from 2017 (labeled 14C, AS3b, AS4e, and AS6b), and (iv) spring fluid from 2016 (labeled BFP16 Spring) (see Table 2 and Fig. 4; see also Fig. S2 in the supplemental material). Cryoconite sediment, aufeis, and spring fluid samples for DNA extraction were filtered through 0.22-$\mu$m-pore-size Luer-Lok Sterivex filters (EMD Millipore; Darmstadt, Germany), which were then capped and kept on ice in the field until they were returned to the laboratory, where they were stored at −20°C until extraction. Aufeis surface scrapings (mineral precipitate samples) from the 2017 field campaign were collected using sterile and field-washed transfer pipettes, and up to 0.5 g of material was mixed in ZR BashingBead lysis tubes containing 750 ml of DNA/RNA Shield (Zymo Research Corp.). Samples were shaken and kept at 4°C until they were returned to the laboratory, where they were then stored at −80°C until DNA/RNA extraction could be performed.

**DNA extraction and metagenomic library preparation.** DNA extraction was performed with 2014 melt pool and aufeis samples and 2016 spring fluid samples by the use of a PowerWater Sterivex DNA isolation kit (MO BIO Laboratories, USA), and DNA extraction of 2017 surface precipitate samples was performed using a ZymoBIOMICS DNA/RNA minikit (Zymo Research Corp.). All extractions were performed according to the instructions of the manufacturers. Concentrations of extracted DNA were determined using a Qubit 2.0 fluorometer (Thermo Fisher Scientific, Chino, CA, USA) and the Qubit double-stranded DNA (dsDNA) high-sensitivity (HS) assay (Life Technologies, Carlsbad, CA, USA).

Genomic DNA was first cleaned using Kapa Pure beads (Kapa Biosystems Inc., Wilmington, MA, USA) at a final concentration of 0.8× (vol/vol) prior to library preparation. Two metagenomic sequencing runs were completed for the included sample set. The first (including samples BFP14 A4b, A6, and M4b) were normalized to 1 ng of DNA in 35 $\mu$l of molecular-biology-grade water and were prepared using a Kapa HyperPlus kit (Kapa Biosystems Inc., Wilmington, MA, USA) and Kapa WGS adapters. Fragmentation and adapter ligation were performed according to the manufacturer's instructions. The prepared samples were subjected to quality checking on an Agilent 2100 Bioanalyzer (Agilent Genomics, Santa Clara, CA,

USA) and submitted to the Duke Center for Genomic and Computational Biology (Duke University, Durham, NC, USA). Final pooled libraries were run on an Illumina HiSeq 4000 system (Illumina, San Diego, CA, USA) using PE150 chemistry. The second sequencing run (which included samples BFP14 M2; BFP16 Spring; and BFP17 AS3b, AS4e, AS6b, and 14C) was prepared using a Nextera XT DNA library preparation kit (Illumina, San Diego, CA, USA) in conjunction with Nextera XT V2 indices. Libraries were hand normalized and quality checked on an Agilent 2100 Bioanalyzer (Agilent Genomics, Santa Clara, CA, USA). Samples were submitted to the Duke Center for Genomic and Computational Biology (Duke University, Durham, NC, USA). These libraries were sequenced on an Illumina HiSeq 2500 Rapid Run system using PE250 chemistry.

**Metagenomic sequencing assembly and analysis.** Processing of metagenomic sequencing was performed using the Summit High Performance Cluster (HPC) located at the University of Colorado Boulder (Boulder, CO, USA). Assembly of whole-genome sequencing reads was performed by the use of a modified Joint Genome Institute (JGI; Walnut Creek, CA, USA) metagenomics workflow. We began by using BBDuk of the BBMap/BBTools (v37.56; 66) suite to trim out Illumina adapters using the built-in BBMap adapters database. Trimmed sequences were checked for quality using FastQC v11.5 (67) to ensure that adapters were removed successfully. Trimmed paired-end sequences were then joined using the PEAR package (68) v0.9.10 and a *P* value of 0.001. Both forward (R1) and reverse (R2) adapter-trimmed files were then processed using a BBTools script ('repair.sh') to ensure that the read pairs were in the correct order. This script is designed to reorder paired reads that have become disorganized, which can often occur during trimming. The BFC (Bloom Filter Correction) software tool was then used to error-correct paired sequence reads (69). Reads were again processed with the BBTools repair.sh script to ensure proper read pairing and order. Next, Megahit v1.1.2 (70) was used to generate a coassembly as well as individual sample assemblies from the quality controlled reads. After assembly, the coassembly was processed with 'anvi-script-reformat-fasta' for input into Anvi'o, and contigs with lengths below 2,500 bp were removed. Bowtie2 v2.3.0 (71) was then used to map individual sample sequence reads onto the coassembly, a procedure needed for the Anvi'o pipeline. Anvi'o v4 (38) was then used to first characterize and then bin coassembled contigs into metagenome-assembled genomes (MAGs). The taxonomic classifier Centrifuge v1.0.2-beta (72) was used to provide a preliminary taxonomy of the coassembly contigs, useful for Anvi'o binning. The NCBI Clusters of Orthologous Groups (COGs; 73, 74) database was used within Anvi'o to predict COGs present within the coassembly, and CONCOCT v1.0 (37) was employed to create genome bins using a combination of sequence coverage and composition of our assembled contigs. Anvi'o was then used to visualize and manually curate putative bins.

Anvi'o refined bins were quality checked via CheckM v1.0.11 (75), where a total of 31 bins were selected as medium-quality MAGs that followed criteria of >50% completeness and <10% redundancy (contamination). The MIMAG (Minimum Information about Metagenome-Assembled Genome) standards (39) developed by the Genomic Standards Consortium (GSC) were used as a guideline for determining which MAGs to include in the manuscript and the included reporting statistics (Table 1; see also Fig. 3). MAGs that met these criteria were then renamed using numbers from 1 to 31 in the order of highest percentage completeness to lowest (Table 1). CheckM relies on other software, including pplacer v1.1.alpha17 (76) for phylogenetic tree placement, prodigal v2.6.3 (77) for gene translation and initiation site prediction, and HMMER v3.1b2 (78) for analysis of sequence data by the use of hidden Markov models. Taxonomic classification of MAGs was performed by running assembled contigs through Genome Taxonomy Database toolkit (GTDB-Tk) software (v1.1.0; 79) and confirming the output with MiGA (Microbial Genomes Atlas; 80). GTDB-Tk has a number of dependencies as well, including the database GTDB (81), pplacer (76), FastANI (82), Prodigal (83), FastTree 2 (84), the "multiple segment Veterbi" (MSV) algorithm (78), and Mash (85). Gene annotations for MAGs and individual site assemblies were generated using Prodigal v2.6.3 (83). Finally, KofamScan v1.2.0 (86) was used to assign KEGG Ortholog (KO) classifications to amino acid annotations, and KEGG Decoder v1.0.10 (87) was used to place KOs into pathways and generate our metabolic pathway heat maps. The scripts used in the processing of sequencing data (up to the point of Megahit-generated contigs) are publicly available at https://doi .org/10.5281/zenodo.1302787.

**Data accessibility.** Raw metagenomic sequencing data are available at the NCBI sequencing read archive (SRA) under accession numbers SRR10066347, SRR10066348, SRR10066349, SRR10066350, SRR10066351, SRR10066352, SRR10066353, SRR10066354, SRR10066355 (https://www.ncbi.nlm.nih.gov/ sra). Assembled MAGs and individual sample metagenomes are provided via FigShare under the following URL: https://doi.org/10.6084/m9.figshare.9767564. Additionally, the bioinformatics workflow up to the point of assembled contigs can be found under the following URL: https://doi.org/10.5281/ zenodo.1302787.

## SUPPLEMENTAL MATERIAL

Supplemental material is available online only.

**FIG S1**, PDF file, 0.2 MB.

**FIG S2**, PDF file, 0.1 MB.

**FIG S3**, PDF file, 2.3 MB.

## ACKNOWLEDGMENTS

Field logistics and travel to and from Borup Fiord Pass were supported through the Polar Continental Shelf Program (PCSP) of Natural Resources Canada (NRCan). J.R.S. and

S.E.G. thank Karsten Piepjohn and the Federal Institute for Geosciences and Natural Resources at BGR, Hanover, Germany, for support during the 2017 field campaign of the Circum-Arctic Structural Events-19 (CASE-19) expedition.

B.W.S. is employed by UES, Inc., which had no involvement in and made no contribution to the presented work. We declare that the research was conducted in the absence of any commercial or financial relationships that could be construed as a potential conflict of interest.

A.S.T., J.R.S., and S.E.G. led the design of the study. Fieldwork was conducted by C.B.T., G.E.L., S.E.G., A.S.T., and J.R.S. Laboratory work was performed by C.B.T. and G.E.L., and bioinformatic analysis was done by C.B.T. and B.W.S. All of us interpreted results. C.B.T. is the primary author of the manuscript with contributions and guidance from all of us.

This research was funded by a grant (NNX13AJ32G) from the NASA Exobiology and Evolutionary Biology Program to A.S.T. and J.R.S. This work was also supported by NASA Astrobiology Institute Rock Powered Life Grant NNA15BB02A (C.B.T., A.S.T., and J.R.S.) and by funding from the Geological Survey of Canada. B.W.S. was supported by the Sloan Foundation grant G-2017-9853. C.B.T. is currently supported by the Helmholtz Recruiting Initiative (award number I-044-16-01 to Liane G. Benning).

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
