## [Reviewer comments · mSystems]

Microbial Metabolic Redundancy is a Key Mechanism in a Sulfur-Rich Glacial Ecosystem

Christopher Trivedi, Blake Stamps, Graham Lau, Stephen Grasby, Alexis Templeton, and John Spear

Corresponding Author(s): John Spear, Colorado School of Mines

Review Timeline:

Submission Date:	June 5, 2020
Editorial Decision:	July 7, 2020
Revision Received:	July 15, 2020
Accepted:	July 17, 2020

Editor: Thulani Makhalanyane

Reviewer(s): Disclosure of reviewer identity is with reference to reviewer comments included in decision letter(s). The following individuals involved in review of your submission have agreed to reveal their identity: Marc Warwick Van Goethem (Reviewer #1)

Transaction Report:

DOI: <https://doi.org/10.1128/mSystems.00504-20>

Comment to Editor

Dear Dr. Makhalanyane,

Response to reviewers on “Metagenomic Insights into Microbial Metabolisms of a Sulfur-Influenced Glacial Ecosystem”

We would like to thank you for receiving our manuscript and the reviewers for their helpful comments and suggestions. We have decided to resubmit our manuscript to *mSystems* per your suggestion and feel that it is now a much stronger manuscript. We have addressed all concerns that were raised, and the revised manuscript has been significantly improved thanks to the suggestions of the reviewers. Thank you for your time on this matter.

Reviewer 1

Review for the manuscript titled: “Metagenomic Insights into Microbial Metabolisms of a Sulfur-Influenced Glacial Ecosystem”

In this manuscript, Trivedi et al, describe the genomic attributes of metagenome-assembled genomes (MAGs) reconstructed from microbial communities collected from various sulfur-rich niches in the low-temperature high Arctic. The authors queried 31 medium- to high-quality MAGs for genes involved in Carbon, Nitrogen and, more importantly, Sulfur cycling. They then discuss the potential implications for sulfur cycling dynamics in this unique environment. The results are interesting and tackle a biogeochemical cycle that is often ignored, thus offering novel insights into microbial ecology in cold regions while certainly building upon taxonomic insights provided by Trivedi et al. (2018).

However, I believe that the authors require additional analyses to reinforce the genetic observations of the MAGs (discussed below). I recommend a major revision before acceptance for publication in *mSystems*.

Major Comments:

The manuscripts primary focus is the analysis of 31 MAGs derived from the sulfur-rich samples. Based on MIMAG standards (Bowers et al., 2017), authors need to report many more statistics than are presented here. Firstly, I don't see evidence of a contamination estimate, only redundancy. What are contamination values or are these the same statistic? If we use the completion and redundancy values, then 9 MAGs are of high-quality, while the remaining 22 are medium-quality. However, the authors still need to report additional statistics such as L50, longest contig, predicted genes per genome, tRNAs and the presence of 5S, 16S and 23S rRNA genes, among others. Combined, these data will inform as to the current accepted quality of each MAG and the reader will be more convinced of their relevance in this system.

The reviewer makes a very important point here. We have rectified our mistake and now made sure our statistical information in Tables 1 and 2 satisfy the MIMAG standards put forth by Bowers et al. and the rest of the Genomic Standards Consortium (GSC).

My second major suggestion is to utilize the available contigs that were used to create the MAGs. Although there is merit in the analysis of genes from the MAGs, I wonder if this could be under-selling the great data that the authors have available. Since not all species are reconstituted as MAGs (according the 16S rRNA data presented in Trivedi et al., 2018) it would be parsimonious to also analyze the gene content of the assembled metagenomes. Although this task is not entirely trivial, it would certainly reinforce the observations made about sulfur cycling derived from the MAGs.

The reviewer makes another excellent point here. We have now updated our dataset to also include the individual assembled sample metagenomes. Additionally, we have re-run our functional analysis to identify the completeness of major metabolic pathways as opposed to searching for individual genes within those pathways, which was previously done. We think this has added significant value to the manuscript and has addressed the concerns of the reviewer.

Minor Comments:

Introduction

Line 75: “implicated in playing a role in” is redundant.

The reviewer makes a very good point. This has been removed. Now on *Line 76*.

Results

Line 147: “MAG 31 had abundant contribution from samples...” Is this based on mapping rates?

Are these the values in Fig. 3A?

The reviewer asks a very good question. I see how this statement is confusing. Yes, the “abundant contribution” is attributed to the read coverage from multiple samples to MAG 31. The values are shown as the monochrome heatmap (with darker boxes meaning more reads from a given sample was recruited to a given MAG) which is now standalone Figure 4. I have clarified the text, which was also requested by Reviewer 2.

Revised discussion: *Line 147*

“MAG read coverage was variable between the different BFP sites (Figure 3). For example, MAG 31 (most closely related to the genus *Flavobacterium*) had abundant read contributions from samples A6 (aufeis), AS3b, AS4e, and AS6b (all mineral precipitate samples), and moderate contributions from A4b and M4b (aufeis and melt pool samples, respectively). However, MAG 31 was also the least complete of the reported MAGs by genome completeness percentage.”

Line 165: “within 17 of 31 of MAGs” should be “within 17 of the 31 MAGs”

I thank the reviewer for pointing this out. We have reworded much of the manuscript and this statement has now been removed.

Lines 176-178: The authors explain that Nitrite reductase genes were more abundant across MAGs but were not especially abundant. This is confusing.

The reviewer makes a very good point. We have updated our analyses and rewritten the results to reflect this.

Revised discussion: *Line 176*

“Fully complete dissimilatory nitrate reduction (to nitrite) pathways were present in a number of MAGs (12, 9, 1, 27, 2, 4, and 15) as well as all but AS4e in the sample metagenomes.”

Line 185-186: “HoxU gene abundance was found in 9 out of 31 MAGs”. This sounds strange.

Thank you for pointing this out. Based on our updated analyses we focus on pathways as opposed to single genes for potential functional capability. The updated text reflects this below.

Revised discussion: *Line 184*

“Many pathways for the potential for aerobic respiration were present as part of the MAGs and sample metagenomes. The mostly likely pathways for oxidative phosphorylation within the BFP samples were via F-Type ATPase, Ubiquinol-cytochrome c reductase, cytochrome c oxidase (cbb3 type), and cytochrome bd complex (Figure 4).”

Lines 186-191: The hopanoid biosynthesis gene section could be removed as it is never referred to again. Alternatively, since hopanoids are a subclass of triterpenoids, i.e. natural products, it may be easier to detect them through antiSMASH which is detected to finding co-localized biosynthetic gene clusters that encode for the production of such molecules.

The reviewer makes a very good point here. We were merely attempting to search for biomarker genes as they could be important for the search for extraterrestrial life. We do agree in this case, as it is not referred to again the manuscript, that it should be removed.

Lines 199-201: By now the authors have listed catalogs of genes for different pathways (which is not always stimulating to read), but this sentence could be trimmed for ease of reading. Perhaps rearranging could help? “Sulfide-quinone reductase (sqr) was only observed in four MAGs (3, 5, 6, and 25), three of which (3, 6, and 25) are Alphaproteobacteria and the other most closely related to the genus Limnobacter (also containing the aforementioned fcc gene)” could be “Sulfide-quinone reductase (sqr) was only observed in three Alphaproteobacteria MAGs (3, 6, and 25) and another most closely

related to the genus *Limnobacter* (also containing the aforementioned *fcc* gene)”. I’m sure this is pedantic, but it bugged me while reading.

We apologize to the reviewer for less than stimulating text in the context of the results and highly appreciate their suggestions to these sentences. We have amended our analysis and changed the text to reflect how are results are presented.

Revised discussion: *Line 190*

“MAGs and sample metagenomes were queried for metabolic pathways and well-known genes that encode for enzymes involved in sulfur-based metabolisms: sulfide oxidation (e.g., *fcc*, *sqr*), sulfur oxidation (*sdo*; sulfur dioxygenase), sulfite dehydrogenase (*sorAB*), and thiosulfate oxidation (*sox* and *tsdA*). Full pathways for sulfide oxidation were found within 11 out of the 31 MAGs and all of the sample metagenomes (Figure 4). All but three MAGs (31; *Flavobacterium*, 1; *Sulfurimonas*, and 9; *Sulfurovum*) were part of the Proteobacteria. The remaining MAGs were classified within the Alpha- and Gammaproteobacterial classes.”

Line 219: “genes are seen as part of the” could be “genes are co-localized as the”

The reviewer suggests a very helpful edit to the paper here. Thank you. We have enhanced our analysis and modified the text to reflect this. The section detailing sulfur metabolism results now starts on *Line 189*.

Line 255: “to carry out this function in these environments” could be more specific as “to oxidize sulfur in these environments”

I agree, and I thank the reviewer for this helpful change to the manuscript and have incorporated it into our text on *Line 239*.

Line 266:267: “*Sulfurimonas* and *Sulfurovum* found in abundance at some sample sites”. Based on Figure 3A (which is a bit challenging to read) I wouldn’t say that MAG1 (*Sulfurimonas*) is abundant at any one site, if I’m interpreting the heat map correctly. Are there mapping data to support this notion?

The reviewer makes a very good point that we don’t actually provide microbial abundance in this paper. My wording for this statement came of as confusing, where I meant to say that the majority of mapped reads for MAG 1 were recruited from site M2. I have updated the text to better reflect this sentiment and to be clearer.

Revised discussion: *Line 252*

“It should also be noted that nitrate reduction was found in both MAGs 1 (*Sulfurimonas*) and 9 (*Sulfurovum*), suggesting that sulfur oxidizing microorganisms (SOMs) such as *Sulfurimonas* and *Sulfurovum* found in higher relative abundance (Trivedi et al., 2018) at some sample sites (e.g., M2 and M4b, respectively) are capable of oxidizing reduced sulfur compounds and utilizing nitrate as a terminal electron acceptor.”

Line 280: “S0” is this correct? I’ve also seen “S0” elsewhere. Also, could the authors speculate whether trace gas scavenging could also supplement microbial energy needs in the High Arctic?

Thank you to the reviewer for pointing out this mistake with “S0”, we have fixed it on *Line 282*. Additionally, we have added a few sentences addressing trace gas scavenging at BFP. We appreciate the suggestion and have updated the text as shown below.

Revised discussion: *Line 361*

“The discovery of the hydrogen:quinone oxidoreductase pathway supports data from 2014 (unpublished) where 29 nM of H₂ was measured in one of the melt pools (site M4; 14), indicating that molecular hydrogen could act as one of the trace gas electron donors as an alternative for the oxidation of sulfur.”

Lines 289, 291, 296 and 297: The authors introduce the abbreviation of the reductive acetyl-CoA cycle as rCoA on line 284, but then don’t use it.

The reviewer makes a very good point, that we didn’t even make use of our abbreviation after spending time to define it. We have made the changes and now refer to it as the Wood-Ljungdahl pathway throughout the manuscript.

Line 295-296: “The rCoA cycle requires anoxic conditions which is contrary to the oxygen-rich surface sampling conditions at BFP”. Although the authors do consider the establishment of anoxic zones, we must remember that this discussion relates to genomic information, and not actual functional measurements. I don’t think that it is implausible that microbes carry genes for different environmental situations that they may be exposed to.

The reviewer makes a very good point here. In fact, it is very likely that microbes carry multiple functional genes to deal with a variety of potential environmental situations. We have amended our analysis our text. I include below our updated speculation about the Wood-Ljungdahl speculation.

Revised discussion: *Line 296*

“The Wood-Ljungdahl pathway was only partially identified in two MAGs (8 and 7) which are taxonomically related to class *Coriobacteriia* and genus *Desulfocapsa*. This agrees with a previous study where it was reported that *Desulfocapsa sulfexigens* was able to thrive on CO₂ as its sole carbon source via the reverse acetyl-CoA (Wood-Ljungdahl) pathway (44). There was another MAG (12) classified as *Desulfocapsa*, not as complete as MAG 7 (Table 1), which could explain why the genes for the Wood-Ljungdahl pathway were not observed in our analysis. Additionally, the Wood-Ljungdahl pathway is only partially complete in sample metagenomes M2 and AS3b, of which M2 had a larger relative abundance of *Desulfocapsa* in the 2018 reported data.”

Line 346: “The lack of soxCD genes” could this be a database issue? Perhaps it is because the genomes aren’t complete? Can you assign any soxCD genes in the metagenomes to any sulfur-oxidizing microorganisms? This would validate the assumptions that arise from the absence of these genes. Could the authors make a comparison to complete genomes of these genera so that the absence of genes can be better contextualized?

The reviewer brings up a very good point here, and notes a few key reasons why these genes could be missing from our analysis. Based on feedback from the reviewer we have now performed additional analyses to look at metabolic pathway completion rather than singular genes. This has shown us that thiosulfate oxidation, where the *soxCD* genes should be present is in fact complete for these MAGs. While *soxC* is missing in MAG 19 (*Thiobacillus*), and a known SOM, we believe this is because the genome is incomplete. We have greatly modified our text to reflect this new information.

Revised discussion: *Line 347*

“The soxCD enzyme complex has been shown to be responsible for the oxidation of sulfur to thiosulfate, and in organisms that lack this complex the sulfur is either stored inside the cell or excreted (55). The lack of the *soxC* gene in BFP MAG 19 from the known sulfur-oxidizer *Thiobacillus* is an interesting finding. In fact, this may be one possible biological explanation for the abundance of S⁰ precipitated across the surface of BFP. However, it may be more likely that the *soxC* gene is missing as the genome for MAG 19 is only 72% complete (Table 1).”

Lines 386-387: Once again I believe that the analysis of functional genes from the metagenomes would really support the argument for functional redundancy at BFP. Clearly the MAGs are incomplete and only capture a subset of the microbial community at this important location. Analyzing the functional genes from all sequenced community members would reveal how many members have the capacity to contribute to sulfur cycling.

We thank the reviewer for this great suggestion. We have taken this advice into account and now also analyzed the metagenomes from each site sample to look at metabolic pathways and how they compare with the co-assembled MAGs. This has helped strengthen our story greatly and we have updated the text significantly due to this addition.

Line 396: “facilitated from above by the input of new microbiota” please remove the “from above” part.

We thank the reviewer for pointing out this text. We have made the suggested change on *Line 415*.

Line 492: “MAGs with 50% completeness and 10% redundancy”. I’m not sure where these thresholds were derived from, but the correct citation should be Bowers et al., 2017 (Minimum information about a single amplified genome (MISAG) and a metagenome-assembled genome (MIMAG) of bacteria and archaea). This will inform the authors as to which other stats to include in Table 1.

The reviewer points out confusing terminology, which we agree should be clarified. We adopted the terminology presented by the developers of Anvi'o (<http://merenlab.org/2016/06/09/assessing-completion-and-contamination-of-MAGs/>). We also think the term 'contamination' to be unnecessarily negative and prefer to use the term 'redundancy' to reflect the presence of single-copy marker genes in MAGs. We have updated our text to better explain this.

Revised discussion: *Line 502*

“Anvi'o refined bins were quality checked via CheckM v1.0.11 (75) where a total of 31 bins were selected as medium quality MAGs (Metagenome Assembled Genomes) that followed the > 50% completeness and < 10% redundancy (contamination) criteria. The MIMAG (Minimum Information about Metagenome-Assembled Genome) standards (39) developed by the Genomic Standards Consortium (GSC) were used as a guideline for which MAGs we've chosen to include in this manuscript, and the included reporting statistics (Table 1 and Figure 3).”

Line 498: The authors use CheckM for phylogenetic placement, which is probably correct. I do wonder if using a dedicated MAG taxonomy tool such as MiGA (<http://microbial-genomes.org/>) would be beneficial or corroborate these results.

The reviewer makes a very good point. We have now extended how we have done our taxonomy calling and have processed our MAGs through the GTDB and checked those results with MiGA. These updated taxa can now be found in Table 1. While some taxonomical calls have changed slightly, all high levels (Phyla, etc.) are still the same except for the Betaproteobacteria, which have now been removed as an individual phylum and placed under the Gammaproteobacteria. These new taxonomies have been updated throughout the manuscript and we thank the reviewer for their helpful suggestion. We have also amended our methods to reflect this helpful suggestion.

Revised discussion: *Line 533*

“Taxonomic classification of MAGs was determined by running assembled contigs through the Genome Taxonomy Database toolkit (GTDB-Tk) software (v1.1.0; 79) and confirming the output with MiGA (Microbial Genomes Atlas; 80). GTDB-Tk has a number of dependencies as well, including the database GTDB (81), pplacer (76), FastANI (82), Prodigal (83), FastTree 2 (84), the “multiple segment Veterbi” (MSV) algorithm (78), and Mash (85).”

Reviewer 2

“Metagenomic Insights into Microbial Metabolisms of a Sulfur-Influenced Glacial Ecosystem”

Authors: C. Trivedi, B. Stamps, G. Lau, S. Grasby, A. Templeton, J. Spear

Journal: mSystems

Recommendation: needs major revisions

In this study, the authors analyzed 31 MAGs from sedimented material, glacial fluids, and surface precipitates obtained from a site in Canadian Arctic. The manuscript is focused primarily on questions regarding the sulfur cycling. This study shows that sulfur metabolism, especially S oxidation is widespread at this site across 4 subclasses of Proteobacteria. This expands substantially on previous work from this site that concluded that Gamma- and Epsilonproteobacteria were largely responsible for sulfur oxidation. The authors conclude that functional redundancy may be key in polar, ephemeral sulfur-based environments. In addition to S metabolism genes, the authors also queried the MAGs for N- and C-cycling, including carbon fixation pathways. Several genera were identified in this study that are rarely identified in polar environments and, in addition, were missing from an earlier 16S rRNA study at this site. To characterize the microbial metabolisms even further, metatranscriptomic sequencing is suggested to complement this metagenomic study.

This manuscript presents an impressive metagenomic dataset from a very intriguing arctic environment. The analysis of the 31 MAGs reveals important new understanding of S-cycling in cold, freshwater systems that may also provide insight into fundamental astrobiological questions, especially on icy worlds (e.g., Europa). **The writing, however, needs to be tightened significantly. Far too many sentences are awkward, inconsistent, vague, unclear, or incorrect. Here, I give many, but certainly not all, examples from just the first approximately 200 lines of text. Similar issues exist throughout the rest of the manuscript and captions as well.** I start with a few general observations and then provide line-specific comments:

The authors need be consistent in their use of Borup Fjord Pass (BFP). In the text, BFP is defined as an environment, a system, an ecosystem, a glacial system, a site, and a valley. I assume given its name that it is also a pass. Also, nearly the exact same text is used 4 times to introduce BFP, and this abbreviation is defined 4 times (Lines 29, 63, 245, 293).

I encourage the authors to not overuse the word “important”. It is used to highlight sulfur metabolism, other pathways, sulfur redox, SRMs, system classification, carbon utilization, and survival mechanisms. **The reviewer makes a very valid point, we have made changes throughout to address this.**

Sometimes the word ‘data’ is used incorrectly as a singular noun, sometimes it is used correctly as a plural noun. The use of ‘within’ throughout the text seems awkward at times. Also, the use of ‘instance’ is awkward at times. **The reviewer makes another great point about our overuse of certain words. We have changed a substantial portion of the text to address many of the readability problems.**

We would like to thank the reviewer for going through the first part of the manuscript for sentence structure and grammatical errors. We realize that poorly written sentences can ruin a paper, and we appreciate your attention to detail and making this a better manuscript. We have addressed all of your line-by-line concerns and made substantial changes to the rest of the paper in order to make it less awkward and a more enjoyable read. Thank you kindly for your time.

Lines

82-84 What does it mean to produce ‘a metagenome in the context of geochemical data to constrain the bioenergetics of microbial metabolism from a mound ...’?

The reviewer makes a very good point here, this is a confusing and poorly written sentence. We have rewritten the text to clarify our statement.

Revised discussion: *Line 83*

“Wright et al. (9) produced one metagenome from a mound of elemental sulfur sampled at the site in 2012. They used this metagenome to detail the bioenergetics of microbial metabolisms and found that at least in surface mineral deposits, sulfur oxidation was likely the dominant metabolism present among the *Epsilonproteobacteria* (now *Campylobacteria*).”

86-87 How can a 16S study reveal an ‘active’ assemblage? I think it is misleading to assume activity here.

The reviewer makes a very good point here. We should have worded this differently. We have removed the portion about the microorganisms being “active”.

Revised discussion: *Line 86*

“An exhaustive 16S rRNA gene sequencing study conducted on samples collected in 2014-2017 revealed a diverse assemblage of both autotrophic and heterotrophic microorganisms in melt pools, auefis, spring fluid, and surface mineral deposits that persist over multiple years, contributing to a basal community present in the system regardless of site or material type (14).”

105-106 Inorganic carbon is invoked as an electron donor here, but apparently not methane, because that is listed separately. Explain more clearly.

The reviewer makes a very good point here. We are not sure why we made the separation between organic carbon and methane in this situation. We have amended the text to make this statement clearer.

Revised discussion: *Line 103*

“Likewise, SRMs can utilize a range of electron donors, including organic carbon, inorganic carbon, hydrogen, metallic iron (26), and can conversely utilize heavy metals such as uranium as

electron acceptors, where the soluble U(VI) is converted to the insoluble U(IV) under anoxic conditions (27, 28).”

107 U(VI) is listed here as a potential electron donor.

We thank the reviewer for pointing this out. You are correct that SRMs reduce U(VI) in this case, using it as an electron acceptor. We have modified our text to reflect this oversight.

Revised discussion: *Line 103*

“Likewise, SRMs can utilize a range of electron donors, including organic carbon, inorganic carbon, hydrogen, metallic iron (26), and can conversely utilize heavy metals such as uranium as electron acceptors, where the soluble U(VI) is converted to the insoluble U(IV) under anoxic conditions (27, 28).”

115-117 How will a better classification of these systems help?

The reviewer points out a weak statement on our part. We have modified the text to give more weight to the statement and explain why classifying ecosystems of this nature is helpful to microbial ecology and astrobiology.

Revised discussion: *Line 112*

“It is important that we better classify these systems as they are key to our understanding of how life can adapt to potentially adverse conditions, such as low-temperatures, highly sulfidic conditions, and low carbon and nutrient levels. Furthermore, these microbial adaptations may also be able to inform us about where to search on worlds outside of Earth and the types of microorganisms to search for. One example where this research is applicable is Europa, where we know that low-temperature, sulfur-rich conditions exist (34–36).”

117-120 A very wordy and awkward sentence, but also, are there no more recent, updated references about S on Europa than this 20+ year-old study?

The reviewer makes a very good point, however, to date, the Carlson paper is still one of the most relevant on the topic. This is because at this point we can only speculate about the source and types of sulfur on Europa. Until the Europa Clipper and hopeful Europa Lander missions we won't have a definitive answer on these two questions. Regardless of this fact, we have modified the text to reflect these points and clarify our statements on the matter.

Revised discussion: *Line 115*

“Furthermore, these microbial adaptations may also be able to inform us about where to search on worlds outside of Earth and the types of microorganisms to search for. One example where this research is applicable is Europa, where we know that low-temperature, sulfur-rich conditions exist (34–36).”

125 How can you tell from MAGs which metabolisms are ‘dominant’?

The reviewer notes a confusing sentence. We have clarified this sentence to remove any idea of activity inferred from metagenomic data.

Revised discussion: *Line 121*

“The sequencing data was assembled and binned into Metagenome Assembled Genomes (MAGs) and sample metagenomes (metagenome assemblies of the site samples themselves) using bioinformatic tools. These assemblies were queried to identify metabolic pathways and their completeness across the BFP samples.”

125-132 I don't think this belongs in the Introduction. It reads like Results and Discussion.

The reviewer makes a good point here. We were merely trying to restate the purpose of our study with a quick summary of our findings and why they are important. We very much appreciate this suggestion by the reviewer, but feel that a recap of the overall aim and results of the story can be an important part of the introduction here. We have shortened the text, but have decided to leave in the main findings and results for the time being.

127 I'm not familiar with a 'taxonomic genome'. What other kind of genome is there? Also, what is 'it' referring to here?

We thank the reviewer for pointing this out. This is confusing and at best is verbose. We have modified the text to be clearer.

Revised discussion: *Line 125*

“Analysis of the MAGs and metagenomes revealed the presence of sulfur oxidation genes (namely those involved in thiosulfate oxidation) across multiple taxonomic phyla, including those related to organisms where it is not predicted to be metabolically viable. This may indicate a form of functional redundancy present in the BFP system where organisms from other Phyla are able to take advantage of the abundance of reduced sulfur for metabolic processes.”

140-142 At the core, this says that 'after assembly, bins were identified after binning'.

The reviewer points out a very obvious mistake. We have modified the text to reflect what we were trying to say.

Revised discussion: *Line 141*

“After assembly, a total of 166 bins were identified using CONCOCT (37) and manual refinement within Anvi'o (38).”

146-149 This is awkward. How can a MAG have contributions from samples?

The reviewer brings up a very good point. This statement as presented doesn't make any sense. What is meant here is that each MAG is made up of reads from various BFP samples. I have

modified the text to make it clearer that reads from each sample are contributing to MAG construction.

Revised discussion: *Line 147*

“MAG read coverage was variable between the different BFP sites (Figure 3). For example, MAG 31 (most closely related to the genus *Flavobacterium*) had abundant read contributions from samples A6 (aufeis), AS3b, AS4e, and AS6b (all mineral precipitate samples), and moderate contributions from A4b and M4b (aufeis and melt pool samples, respectively). However, MAG 31 was also the least complete of the reported MAGs by genome completeness percentage.”

149-150 A MAG can't 'have' a completion. I think what is meant is that MAG 31 was the least complete. Better yet, given the percentage completion as well.

The reviewer is absolutely correct. We worded this poorly and have rectified the mistake by modifying the text.

Revised discussion: *Line 151*

“However, MAG 31 was also the least complete of the reported MAGs by genome completeness percentage.”

151-159 This paragraph is very poorly written. 'MAGs identified within the domain' sounds awkward. I think what is meant is that all MAGs were bacterial. Why say 'six and six, respectively'? I don't think that the Alphabacteria 'contained' MAGs, but rather that 5 MAGs were from the Alphas. *Desulfocapsa* is a genus, not a genera. Be specific about 'highly complete'.

We agree with the reviewer that this is a poorly written paragraph. We have reworded it to be clearer and more concise.

Revised discussion: *Line 153*

“All MAGs were taxonomically classified as Bacteria, the majority of which (17 of 31) within the phylum *Proteobacteria* (Table 1). Of these, eight belong to the *Gammaproteobacteria* (MAGs 2, 4, 11, 15, 16, 23, 27, & 29), five to the *Betaproteobacteria* (MAGs 5, 6, 14, 19, & 20), four to the *Alphaproteobacteria* (MAGs 3, 10, 24, & 25), two to the *Desulfobacterota* (MAGs 7, & 12) one to *Desulfuromonadota* (MAG 21), and two to the *Campylobacteria* (MAGs 1 and 9). Each class had at least one MAG considered highly complete (>90%) with low redundancy (<10%; Table 1).”

163 I think that RuBisCo is generally referred to as the key enzyme, not the pathway.

The reviewer is correct and we thank them for pointing this out. We have significantly changed the text and reworded our results section. We highlight the revised text below where we use RuBisCO later on in the discussion section.

Revised discussion: *Line 286*

“This includes the gene that encodes for RuBisCO (ribulose 1,5 biphosphate carboxylase), a key enzyme involved in carbon fixation.”

165 This is the first mention of Fig. 3b, but Fig. 3a hasn't been mentioned yet. In fact, Fig. 3a isn't mentioned until Line 267.

The reviewer makes a very good point and we thank them for pointing it out. Due to our updated analysis we have rewritten large portions of the results and discussion and updated our figures to reflect this as well. Figure 3 has now been separated into two figures and the text has been updated to reflect their correct order.

169 cbbL and cbbM have not yet been defined. And you shouldn't say 'both X or Y', but rather 'both X and Y'.

The reviewer makes a very good point that we had neglected to define these genes. The grammar in which we presented them was poor as well. Based on our updated analysis we have updated our results and changed our text concerning aerobic respiration which is shown below.

Revised discussion: *Line 184*

“Many pathways for the potential for aerobic respiration were present as part of the MAGs and sample metagenomes. The mostly likely pathways for oxidative phosphorylation within the BFP samples were via F-Type ATPase, Ubiquinol-cytochrome *c* reductase, cytochrome *c* oxidase (*cbb3* type), and cytochrome bd complex (Figure 4).”

173 'Genes identified as nap' sounds awkward.

The reviewer is correct, this sounds awkward. As part of our updated pathway analysis we have significantly revised our results and the updated text concerning nitrate reduction is below.

Revised discussion: *Line 176*

“Fully complete dissimilatory nitrate reduction (to nitrite) pathways were present in a number of MAGs (12, 9, 1, 27, 2, 4, and 15) as well as all but AS4e in the sample metagenomes.”

176-178 This sentence doesn't make sense. Also, more than what?

The reviewer makes a very good point at the confusing nature of this sentence. As part of our updated analysis we have rewritten our results concerning the reduction of nitrite and copied it below.

Revised discussion: *Line 178*

“Additionally, three MAGs (12, 27, and 4) contain genes for a complete dissimilatory nitrate to ammonium (DNRA; NO_3^- to NH_4^+) pathway.”

178-179 MAGs don't have potential to reduce. They can encode enzymes that catalyze the reduction perhaps.

The reviewer makes a very good point. We have revised a significant portion of the manuscript and made sure that throughout we indicate the difference between MAGs and genes that encode for enzymes.

181 I don't think that the oxidase was searched, but rather that there was a search for the oxidase which represents a marker.

The reviewer is correct, this was poorly worded. As part of our updated analysis we have rewritten the results section on aerobic respiration which is copied below.

Revised discussion: *Line 184*

“Many pathways for the potential for aerobic respiration were present as part of the MAGs and sample metagenomes. The mostly likely pathways for oxidative phosphorylation within the BFP samples were via F-Type ATPase, Ubiquinol-cytochrome *c* reductase, cytochrome *c* oxidase (*cbb3* type), and cytochrome bd complex (Figure 4).”

185-186 I think you mean ‘hoxU’ and not ‘HoxU’ here. Also, I think you can find a gene, but you can't find gene abundance.

The reviewer is correct, the wording here is confusing. Based on our updated analysis we have approached metabolisms in a more pathway-based way. We have changed the text accordingly and included it below.

Revised discussion: *Line 184*

“Many pathways for the potential for aerobic respiration were present as part of the MAGs and sample metagenomes. The mostly likely pathways for oxidative phosphorylation within the BFP samples were via F-Type ATPase, Ubiquinol-cytochrome *c* reductase, cytochrome *c* oxidase (*cbb3* type), and cytochrome bd complex (Figure 4).”

188 What is meant by ‘used as a diagnostic for biomarker potential’?

The reviewer points out a confusing statement. Based on this comment and another made by Reviewer 1 about the relevance of this section it has been removed to make the story more concise.

188-190 This sentence is a non sequitur. Also, why is it ‘useful in conjunction with metabolisms’? I would think that finding biomarkers would be useful on their own.

We thank the reviewer for pointing this out. As mentioned in the previous comment we have removed this section to clean up the story.

193-194 This says that oxidation and reduction are genes.

The reviewer makes a good catch and points out an obvious mistake on our part. We have changed the text to rectify this.

Revised discussion: *Line 189*

“Of greatest interest for this study were sulfur cycling associated genes (including genes involved in both the oxidation and reduction of sulfur species).”

194-195 Genes are not involved in oxidation, but they encode enzymes that are.

The reviewer points out another area where we need to be more clear about our scientific grammar. We have changed the text to reflect this.

Revised discussion: *Line 190*

“MAGs and sample metagenomes were queried for metabolic pathways and well-known genes that encode for enzymes involved in sulfur-based metabolisms: sulfide oxidation (e.g., *fcc*, *sqr*), sulfur oxidation (*sdo*; sulfur dioxygenase), sulfite dehydrogenase (*sorAB*), and thiosulfate oxidation (*sox* and *tsdA*).”

202 What is meant by ‘genes necessary for sulfite dehydrogenase’? Again, genes encode.

We thank the reviewer for continuing to point out where we have used the term gene improperly. We have modified the text to correct this.

Revised discussion: *Line 197*

“Ten MAGs and nine sample metagenomes contained the gene that encodes for sulfur dioxygenase (*sdo*), which is often responsible for the oxidation of elemental sulfur. Additionally, 15 out of 31 MAGs, and all sample metagenomes had genes necessary to encode for sulfite dehydrogenase (*sorAB*).”

203-205 The Sox system is not a pathway.

The reviewer makes another good point. We need to be more careful about our wording. We have since changed the text to be more clear about this.

Revised discussion: *Line 201*

“The Sox enzyme complex is widely-studied for its ability to facilitate the complete oxidation of thiosulfate ($S_2O_3^{2-}$) to sulfate (SO_4^{2-}).”

I will stop here. I'm not trying to be overly picky, but I think that a very interesting dataset that illuminates some very interesting microbial processes is lost among far too many poorly written sentences.

We want to apologize to the reviewer for the abundance of poorly written sentences and grammatical errors. We also want to thank the reviewer for reading through 200 lines of manuscript and providing detailed and thoughtful comments. We have amended the manuscript significantly and modified the text substantially to be much clearer and hopefully more readable. We thank the reviewer for their consideration and time.

July 7, 2020

Dr. John R. Spear
Colorado School of Mines
Environmental Science and Engineering
1500 Illinois Street
Golden, CO 80401

Re: mSystems00504-20 (Microbial Metabolic Redundancy is a Key Process in a Sulfur-Influenced Glacial Ecosystem)

Dear Dr. John R. Spear:

Thank you for revising your manuscript. You will note that the reviews are rather mixed. Reviewer 2 has provided detailed suggestions on your revised ms. These are mostly grammatical edits which are provided to help improve the precision and clarity of your manuscript. Please take care while revising your manuscript to ensure that all the comments are addressed.

Below you will find the comments of the reviewers.

To submit your modified manuscript, log onto the eJP submission site at <https://msystems.msubmit.net/cgi-bin/main.plex>. If you cannot remember your password, click the "Can't remember your password?" link and follow the instructions on the screen. Go to Author Tasks and click the appropriate manuscript title to begin the resubmission process. The information that you entered when you first submitted the paper will be displayed. Please update the information as necessary. Provide (1) point-by-point responses to the issues raised by the reviewers as file type "Response to Reviewers," not in your cover letter, and (2) a PDF file that indicates the changes from the original submission (by highlighting or underlining the changes) as file type "Marked Up Manuscript - For Review Only."

Due to the SARS-CoV-2 pandemic, our typical 60 day deadline for revisions will not be applied. I hope that you will be able to submit a revised manuscript soon, but want to reassure you that the journal will be flexible in terms of timing, particularly if experimental revisions are needed. When you are ready to resubmit, please know that our staff and Editors are working remotely and handling submissions without delay. If you do not wish to modify the manuscript and prefer to submit it to another journal, please notify me of your decision immediately so that the manuscript may be formally withdrawn from consideration by mSystems.

To avoid unnecessary delay in publication should your modified manuscript be accepted, it is important that all elements you upload meet the technical requirements for production. I strongly recommend that you check your digital images using the Rapid Inspector tool at <http://rapidinspector.cadmus.com/RapidInspector/zmw/>.

Sincerely,

T hulani Makhalanyane

Editor, mSystems

Journals Department
Reviewer comments:

Reviewer #1 (Comments for the Author):

The manuscript has been much improved by the authors by taking on all the issues raised by both reviewers.

Although I remain unconvinced by using "redundancy" over "contamination" I do note that these terms could be synonyms and that this remains a debated topic.

I only noticed an issue on line 247 where "Dissimilatory nitrate reduction to nitrate" occurs, this should be to nitrite.

Reviewer #2 (Comments for the Author):

I reviewed this revised version as well as an earlier version of this manuscript. I maintain that this study reports an impressive metagenomic dataset from a noteworthy arctic environment. The focus is primarily on sulfur cycling, and in the revised version the emphasis is placed on metabolic redundancy (see the new title). Carbon fixation is another focal point. My main criticisms in the first round of reviews was the writing. There were many inconsistencies, false statements, and awkward sentences. The writing has improved, but there are still a lot of passages that should be much clearer, and preferably more quantitative (especially with regard to 'completeness' of genomes and pathways). In other words, the writing gets in the way of very interesting results, and that is a shame. I again provide a list of some of the concerns with the text (not the data).

1. Title: is 'redundancy' a 'process'? In the Abstract, it is referred to as 'mechanism', which may be better.
2. Check text carefully; there are extra spaces, capitalization issues, and other minor mistakes throughout.

3. Line 79: 'biological' is superfluous here.
4. Lines 81-84: It sounds awkward to say 'surveyed ... redox bioenergetics' and 'used this metagenome to detail the bioenergetics'.
5. Line 96: here and elsewhere, don't use '&' in place of 'and'.
6. Lines 102-103: 'autotrophically' is superfluous, because CO₂ can't be fixed heterotrophically. Also, CO₂ can't be fixed 'with a variety of electron acceptors', only with electron donors.
7. Line 124: 'assemblies were queried to identify ... pathways and their completeness' is awkward.
8. Line 126: Can phyla be anything other than 'taxonomic'? How about 'across multiple taxa' or 'across multiple phyla'?
9. Lines 153-158: I'm not sure every group of organisms mentioned here is a 'class'.
10. Lines 162-163: '... potential was inferred by looking at the completeness ...' sounds awkward.
11. Line 171: Here and throughout, what does 'partially complete' mean? 5%? 95%? Can you give numbers? In Line 173 it is at least semi-quantitative, noting <50%.
12. Line 184: 'Many pathways for the potential for aerobic respiration' sounds awkward.
13. Lines 195-196: Would be easier to follow if numbers were in order: 1, 9, 31.
14. Lines 204-212: Same here and elsewhere, easier to follow if numbers were in order.
15. You invoke functional redundancy here, but the detailed discussion of this topic comes only much later. Either discuss functional redundancy related to S here, or indicate that it'll be discussed in detail below.
16. Line 247: this should say 'nitrate reduction to nitrite'.
17. Line 253-254: The abbreviation SOM was already defined in Line 99.
18. Line 259: 'anaerobic denitrification'? As opposed to what, aerobic denitrification, which doesn't make sense?
19. Lines 263-264: Aerobic respiration can't be 'present' in MAGs or metagenomes.
20. Lines 265-268: reductase, oxidase, and bd complex are not pathways.
21. Lines 273-275: This sentence is very awkward.
22. Lines 275-277: do Wright et al. really claim that energy production is from carbon fixation?
23. Lines 277-279: When you say 'sulfide oxidation coupled to CO₂ fixation', you're not suggesting CO₂ as the electron acceptor in sulfide oxidation, right? Aerobic sulfide oxidation is the catabolic process, and CO₂ fixation is the anabolic process, I assume. You should be explicit as to your metagenomic data that support this combo of anabolism and catabolism into one of the most common metabolisms here.
24. Lines 300-302: What is meant by 'there was another MAG ... which could explain why genes ... were not observed'?
25. Lines 315-318: Here you write 'high completion', but then follow it up with 'complete pathways' in the next sentence. Are these S pathways complete or almost complete? If the latter, what genes are missing and how conclusive is their absence?
26. Lines 337-340: 'partially complete' is very vague. As suggested above, please be as quantitative as possible. What is meant by 'fully complete pathways for sulfur dioxygenase and sulfite dehydrogenase'? I don't think you mean pathways for the synthesis of these enzymes, right?
27. Lines 345-346: I don't think you meant to say that the enzymes sulfur oxygenase and sulfite dehydrogenase contribute to the oxidized sulfur in the system, but that is what is says.
28. Lines 353-354: I don't think thermodynamics differentiates between biotic and abiotic versions of the same reaction. That's a kinetic argument.
29. Lines 363-364: Molecular hydrogen (an electron donor) cannot oxidize sulfur.

Comment to Editor

Dear Dr. Makhalanyane,

Response to reviewers on “Microbial Metabolic Redundancy is a Key Process in a Sulfur-Influenced Glacial Ecosystem”

We would like to thank you and the reviewers for receiving our resubmitted manuscript, and for providing additional helpful comments and suggestions in this round of review. We have addressed all concerns that were raised by the reviewers in the second round, especially those concerning awkward sentences and grammar mistakes. We feel the manuscript has been significantly improved thanks to the suggestions and feedback of the reviewers. We thank you for all the time you have given to reviewing our manuscript.

Reviewer 1

The manuscript has been much improved by the authors by taking on all the issues raised by both reviewers.

Although I remain unconvinced by using "redundancy" over "contamination" I do note that these terms could be synonyms and that this remains a debated topic.

We would like to sincerely thank the reviewer for generously donating their time to review this manuscript. Your feedback and insight have been invaluable to making this story much stronger and understandable. We certainly note the preference for the use of “contamination” over “redundancy”, and will take this into consideration for the future. Thank you for understanding and respecting our position on this point.

I only noticed an issue on line 247 where "Dissimilatory nitrate reduction to nitrate" occurs, this should be to nitrite.

We thank the reviewer for pointing out this very obvious mistake and have changed the manuscript accordingly.

Revised discussion: *Line 241*

“Based on metabolic pathway analysis we infer that some organisms from BFP samples may be capable of nitrate/nitrite reduction. Dissimilatory nitrate reduction to nitrite, using genes encoded by *nap* (periplasmic nitrate reductase) and *nar* (nitrate reductase) were found in multiple MAGs...”

Reviewer 2

I reviewed this revised version as well as an earlier version of this manuscript. I maintain that this study reports an impressive metagenomic dataset from a noteworthy arctic environment. The focus is primarily on sulfur cycling, and in the revised version the emphasis is placed on metabolic redundancy (see the new title). Carbon fixation is another focal point. My main criticisms in the first round of reviews was the writing. There were many inconsistencies, false statements, and awkward sentences. The writing has improved, but there are still a lot of passages that should be much clearer, and preferably more quantitative (especially with regard to 'completeness' of genomes and pathways). In other words, the writing gets in the way of very interesting results, and that is a shame. I again provide a list of some of the concerns with the text (not the data).

We would like to thank the reviewer for going through the first part of the manuscript for sentence structure and grammatical errors. We realize that poorly written sentences can ruin a paper, and we appreciate your attention to detail and making this a better manuscript. We have addressed all of your line-by-line concerns and made substantial changes to the rest of the paper in order to make it less awkward and a more enjoyable read. Thank you kindly for your time.

1. Title: is 'redundancy' a 'process'? In the Abstract, it is referred to as 'mechanism', which may be better.

The reviewer makes a very good point about using two different terms to describe functional redundancy. We agree that it is better described as a mechanism and have updated the title to reflect this.

“Updated Title: **Microbial Metabolic Redundancy is a Key Mechanism in a Sulfur-Rich Glacial Ecosystem**”

2. Check text carefully; there are extra spaces, capitalization issues, and other minor mistakes throughout.

We agree with the reviewer that there were a number of small mistakes that needed to be fixed. We have gone through the document with a fine-toothed comb to correct these issues and amended the text to reflect this.

3. Line 79: 'biological' is superfluous here.

We agree with the reviewer on this point and thank them for pointing it out. We have removed it from the text as suggested.

Revised discussion: *Line 78*

“Furthermore, organic matter in shales along this fault may be a source of carbon for subsurface microbial processes such as sulfate reduction (10).”

4. Lines 81-84: It sounds awkward to say 'surveyed ... redox bioenergetics' and 'used this metagenome to detail the bioenergetics'.

We thank the reviewer for pointing this out and agree that these sentences were awkward in the way they were presented. We have modified the text to be clearer on our points concerning the Wright et al. paper.

Revised discussion: *Line 80*

“Previous research on the BFP spring has detailed microbial activity, redox bioenergetics (9), the biomineralization of elemental sulfur (S^0 ; 7, 10, 11), and the cryogenic carbonate vaterite (13). Wright et al. (9) produced one metagenome from a mound of elemental sulfur sampled at the site in 2012. From the metagenomic data the authors determined the potential bioenergetics of microbial metabolisms and found that at least in surface mineral deposits, sulfur oxidation was likely the dominant metabolism present among the *Campylobacteria*.”

5. Line 96: here and elsewhere, don't use '&' in place of 'and'.

We agree with the reviewer and thank them for pointing out this mistake. We have removed all instances (16 in total) of “&” and replaced them with the proper “and”.

6. Lines 102-103: 'autotrophically' is superfluous, because CO₂ can't be fixed heterotrophically. Also, CO₂ can't be fixed 'with a variety of electron acceptors', only with electron donors.

We kindly thank the review for pointing out these mistakes. You are exactly correct in that using the term “autotrophically” is redundant in this case. Furthermore, we appreciate the reviewer pointing out our glaring error of referring to CO₂ fixation having a variety of electron acceptors, when it is common knowledge that CO₂ is the electron acceptor, and the reaction can make use of numerous donors. We have amended the text to reflect these changes.

Revised discussion: *Line 101*

“SOMs are metabolically and phylogenetically diverse (15, 23, 24) and can often fix carbon dioxide (CO₂) using a variety of electron donors (25).”

7. Line 124: 'assemblies were queried to identify ... pathways and their completeness' is awkward.

We thank the reviewer for pointing out this sentence and have modified it in the text to make it less awkward.

Revised discussion: *Line 120*

“These assemblies were used to identify metabolic pathways and their completeness across the BFP samples.”

8. Line 126: Can phyla be anything other than 'taxonomic'? How about 'across multiple taxa' or 'across multiple phyla'?

We agree with the reviewer that this is redundant and appreciate their helpful suggestion. We have modified the text to reflect this change.

Revised discussion: *Line 123*

“Analysis of the MAGs and metagenomes revealed the presence of sulfur oxidation genes (namely those involved in thiosulfate oxidation) across multiple phyla, including those related to organisms where it is not predicted to be metabolically viable.”

9. Lines 153-158: I'm not sure every group of organisms mentioned here is a 'class'.

We thank the reviewer for pointing out this mistake. Previously, before the new GTDB and SILVA classifications all of these MAGs were within Proteobacterial classes, however, now their phylogeny has changed. We have updated the text to be clearer on this point.

Revised discussion: *Line 149*

“All MAGs were taxonomically classified as Bacteria, the majority of which (17 of 31) were within the phylum *Proteobacteria* (Table 1). Of these, eight belonged to the *Gammaproteobacteria* (MAGs 2, 4, 11, 15, 16, 23, 27, and 29), five to the *Betaproteobacteria* (MAGs 5, 6, 14, 19, and 20), four to the *Alphaproteobacteria* (MAGs 3, 10, 24, and 25), two to the *Desulfobacterota* (MAGs 7, and 12), one to *Desulfuromonadota* (MAG 21), and two to the *Campylobacteria* (MAGs 1 and 9). Each of the Proteobacterial classes had at least one MAG considered highly complete (>90%) with low redundancy (<10%; Table 1).”

10. Lines 162-163: '... potential was inferred by looking at the completeness ...' sounds awkward.

We thank the reviewer for pointing this out and have modified the text to sound less awkward.

Revised discussion: *Line 160*

“Carbon fixation potential was inferred by evaluating the completeness of five pathways: the Calvin–Benson–Bassham (CBB), reverse tricarboxylic acid (rTCA), Wood-Ljungdahl, 3-hydroxypropionate, and 4-hydroxybutyrate/3-hydroxypropionate pathways.”

11. Line 171: Here and throughout, what does 'partially complete' mean? 5%? 95%? Can you give numbers? In Line 173 it is at least semi-quantitative, noting <50%.

The reviewer makes a very good point here, partially is a bad word to use in this case. We have tried to be more quantitative throughout the manuscript.

Revised discussion: *Line 166*

“The Wood-Ljungdahl pathway was approximately 70% complete (Figure 4) in MAGs 8 and 7, and sample metagenomes M2 and AS3b.”

12. Line 184: 'Many pathways for the potential for aerobic respiration' sounds awkward.

We agree that this is a very awkward sentence, and thank the reviewer for pointing it out. We have amended the text to fix this.

Revised discussion: *Line 180*

“Multiple genes associated with pathways responsible for aerobic respiration were present as part of the MAGs and sample metagenomes.”

13. Lines 195-196: Would be easier to follow if numbers were in order: 1, 9, 31.

We thank the reviewer for pointing this out, but want to note that this was intended based on the order of MAGs within Figure 4. We thought it would be more informative if MAGs were organized by taxonomy rather than MAG # when viewing metabolic pathway potential. While this does make the story seem disjointed when reading the results, we thought this might be a necessary evil, rather than having to jump around while looking at the figure they are at least presented in order from top to bottom. For this reason, we feel it is best to leave it as-is for now.

14. Lines 204-212: Same here and elsewhere, easier to follow if numbers were in order.

As stated in the response above, we feel Figure 4 is best presented as groups of taxa rather than in order of MAG #. We feel it is easier to follow the figure top-to-bottom in this way. We thank the reviewer for pointing this out, however.

15. You invoke functional redundancy here, but the detailed discussion of this topic comes only much later. Either discuss functional redundancy related to S here, or indicate that it'll be discussed in detail below.

Assuming that the reviewer is pointing to the inference of functional redundancy that was on *Line 244*, we had intended to reiterate our findings in the first paragraph of the discussion. We have since taken the reviewer's advice and removed that sentence from the paragraph as we do go into detail later in the discussion.

16. Line 247: this should say 'nitrate reduction to nitrite'.

We thank the reviewer for pointing this out, and note this was also pointed out by Reviewer 1. We apologize for letting a simple mistake (among others) slip though. We have updated our analyses and rewritten the results to reflect this.

Revised discussion: *Line 241*

“Based on metabolic pathway analysis we infer that some organisms from BFP samples may be capable of nitrate/nitrite reduction. Dissimilatory nitrate reduction to nitrite, using genes encoded

by *nap* (periplasmic nitrate reductase) and *nar* (nitrate reductase) were found in multiple MAGs...”

17. Line 253-254: The abbreviation SOM was already defined in Line 99.

We thank the reviewer for pointing out the duplication of the definition in the manuscript. We have removed it from the text as suggested.

Revised discussion: *Line 247*

18. Line 259: 'anaerobic denitrification'? As opposed to what, aerobic denitrification, which doesn't make sense?

We agree with the reviewer that the term “anaerobic” is redundant in this case and we have removed it from the text as suggested.

Revised discussion: *Line 252*

“However, metagenomic evidence also supports the presence of denitrification as a viable mechanism of low-temperature respiration.”

19. Lines 263-264: Aerobic respiration can't be 'present' in MAGs or metagenomes.

The reviewer makes a good point about a confusingly written sentence. We have rewritten the text to hopefully be clearer.

Revised discussion: *Line 257*

“Genes related to aerobic respiration were also present in abundance across BFP MAGs and sample metagenomes.”

20. Lines 265-268: reductase, oxidase, and bd complex are not pathways.

Again, we thank the reviewer for pointing out confusing wording. We have amended the text to clarify our point.

Revised discussion: *Line 259*

“The majority of other aerobic respiration pathways are mostly complete within MAGs related to the *Proteobacteria*, including those that use ubiquinol-cytochrome *c* reductase, cytochrome *c* oxidase (*cbb3* type), and cytochrome bd complex enzymes. These pathways are mostly complete across all sample metagenomes as well.”

21. Lines 273-275: This sentence is very awkward.

We thank the reviewer for pointing this out and have amended the text to be clearer.

Revised discussion: *Line 266*

“Understanding the utilization of carbon in microbial growth dynamics at the BFP site is vital to determining how carbon might enter the system.”

22. Lines 275-277: do Wright et al. really claim that energy production is from carbon fixation?

We thank the reviewer for pointing this out and have amended the text to better reflect the results of Wright et al.

Revised discussion: *Line 269*

“Previous research by Wright et al. (9) indicated that the most likely form of energy production in the system is via aerobic oxidation of S⁰ and that chemolithoautotrophy is the main form of primary production at the site.”

23. Lines 277-279: When you say 'sulfide oxidation coupled to CO₂ fixation', you're not suggesting CO₂ as the electron acceptor in sulfide oxidation, right? Aerobic sulfide oxidation is the catabolic process, and CO₂ fixation is the anabolic process, I assume. You should be explicit as to your metagenomic data that support this combo of anabolism and catabolism into one of the most common metabolisms here.

Again, we thank the reviewer for pointing out a potentially confusing sentence. You are correct that we are not suggesting that CO₂ is the EA in this process, but we can see how our text would suggest this. We have modified the manuscript to be more clear on this point.

Revised discussion: *Line 271*

“Similar to Wright et al., our metagenomic data support the finding that CO₂ fixation rather than the direct oxidation of organic carbon is likely one of the most common metabolisms present at BFP.”

24. Lines 300-302: What is meant by 'there was another MAG ... which could explain why genes ... were not observed'?

We agree with the reviewer that this sentence is poorly worded, and thank them for pointing it out. We have amended the text to better explain.

Revised discussion: *Line 295*

“MAG 12 was also classified as *Desulfocapsa*, however, was not as complete as MAG 7 (84.9% vs 92.8%, respectively; Table 1). This small difference in genome completion might explain why genes for the Wood-Ljungdahl pathway were found in one MAG (7) but not the other (MAG 12).”

25. Lines 315-318: Here you write 'high completion', but then follow it up with 'complete pathways' in the next sentence. Are these S pathways complete or almost complete? If the latter, what genes are missing and how conclusive is their absence?

We agree that this sentence was confusing. We have modified the sentence slightly to show that the pathways we referred to were all complete in terms of presence/absence of the genes that were queried.

Revised discussion: *Line 311*

“One organism, showing full completion of multiple sulfur cycling pathways was *Loktanella* (MAG 10). This MAG shows complete pathways for sulfide oxidation, sulfite oxidation, two forms of thiosulfate oxidation, and (reversible) dissimilatory sulfate reduction.”

26. Lines 337-340: 'partially complete' is very vague. As suggested above, please be as quantitative as possible. What is meant by 'fully complete pathways for sulfur dioxygenase and sulfite dehydrogenase'? I don't think you mean pathways for the synthesis of these enzymes, right?

We thank the reviewer for pointing this out again, and apologize for our use of vague, un-quantitative words. We have updated the text to be more specific about pathway completion (based on the number of genes present or absent) and what these pathways are referring to.

Revised discussion: *Line 334*

“Our *Herminiimonas* MAG (14) contained a partially complete (<50%) thiosulfate oxidation pathway, a fully complete alternative thiosulfate oxidation pathway, and genes that encode for sulfur dioxygenase (*sdo*) and sulfite dehydrogenase (*sorB*), supporting the *in vitro* work by Koh et al. suggesting that *Herminiimonas* has the ability to oxidize sulfur.”

27. Lines 345-346: I don't think you meant to say that the enzymes sulfur oxygenase and sulfite dehydrogenase contribute to the oxidized sulfur in the system, but that is what is says.

We thank the reviewer for pointing this out. We certainly did not mean the genes themselves, rather the organism – in this case *Rhodoferrax*. We have modified the text to better reflect what we were trying to say about the presence of these two genes found in genomic data from BFP.

Revised discussion: *Line 343*

“While not present in any great amount it is possible that *Rhodoferrax* sp. contribute toward the total amount of oxidized sulfur species in the system.”

28. Lines 353-354: I don't think thermodynamics differentiates between biotic and abiotic versions of the same reaction. That's a kinetic argument.

We agree with the reviewer that the thermodynamics do not differentiate between biotic and abiotic forms of the same reaction, however, we were attempting to speculate at the large accumulation and persistence of elemental sulfur at the site. We have revised the language slightly to attempt to provide some further context for kinetic arguments in support of biotic production and/or use of S^0 at this site.

Revised discussion: *Line 351*

“Abiotic sulfur oxidation is predicted to occur at this site based on thermodynamics (9), however, at such low temperatures, this process may be kinetically slow without biological catalysis (and this may explain the large abundance and year-over-year persistence of elemental sulfur at the site).”

29. Lines 363-364: Molecular hydrogen (an electron donor) cannot oxidize sulfur.

We thank the reviewer for pointing this out. We can see how our language was misleading for what we intended to indicate. We were not suggesting that H_2 can oxidize sulfur, but that the presence of high levels of H_2 might serve as an alternate pathway for energy generation in the absence of the ability to oxidize sulfur. We have modified the text to be clearer about this point.

Revised discussion: *Line 360*

“The discovery of the hydrogen:quinone oxidoreductase pathway, a metabolic pathway that uses molecular hydrogen as an electron donor for the reduction of quinone, is intriguing in the face of data collected in 2014 (unpublished), where 29 nM of H_2 was measured in one of the melt pools (site M4; 14). This indicates that molecular hydrogen oxidation through this pathway could act as an alternative to the oxidation of sulfur.”

July 17, 2020

Dr. John R. Spear
Colorado School of Mines
Environmental Science and Engineering
1500 Illinois Street
Golden, CO 80401

Re: mSystems00504-20R1 (Microbial Metabolic Redundancy is a Key Mechanism in a Sulfur-Rich Glacial Ecosystem)

Dear Dr. John R. Spear:

Your manuscript has been accepted, and I am forwarding it to the ASM Journals Department for publication. For your reference, ASM Journals' address is given below. Before it can be scheduled for publication, your manuscript will be checked by the mSystems senior production editor, Ellie Ghatineh, to make sure that all elements meet the technical requirements for publication. She will contact you if anything needs to be revised before copyediting and production can begin. Otherwise, you will be notified when your proofs are ready to be viewed.

Sincerely,

T hulani Makhalanyane
Editor, mSystems

Journals Department
Figure S2: Accept

Figure S1: Accept

Figure S3: Accept